



# Intertidal spring discharge to a coastal ecosystem and impacts of climate change on future groundwater temperature: A multi-method investigation

Jason J. KarisAllen[1], Aaron A. Mohammed[1,2], Joseph J. Tamborski[3], Rob C. Jamieson[1], Serban Danielescu[4,5], Barret L. Kurylyk[1]

[1]Department of Civil and Resource Engineering and Centre for Water Resources Studies, Dalhousie University, Halifax, B3H 4R2, Canada
[2]Department of Earth and Planetary Sciences, McGill University, Montreal, H3A 0E8, Canada
[3]Department of Ocean and Earth Sciences, Old Dominion University, Norfolk, VA, 23529, USA
[4]Water Science and Technology Directorate, Environment and Climate Change Canada, Burlington, ON L7S 1A1, Canada
[5]Agriculture and Agri-Food Canada, Fredericton Research and Development Centre, Fredericton, New Brunswick E3B 4Z7, Canada

*Correspondence to*: Barret L. Kurylyk (barret.kurylyk@dal.ca)

## Abstract

In inland settings, groundwater discharge is known to thermally modulate receiving surface water bodies and provide localized thermal refuges; however, the thermal influence of intertidal springs on coastal waters and the thermal sensitivity of these springs to climate change are not well studied. We addressed this knowledge gap with a field- and model-based study of a threatened coastal lagoon ecosystem in south-eastern Canada. We paired *in-situ* thermal and hydrologic monitoring with analyses of drone-based thermal imagery to estimate the discharge to the lagoon from intertidal springs and groundwater-dominated streams in summer 2020. Results, which were generally supported by independent radon-based groundwater discharge estimates, revealed that the combined summertime spring inflows (0.047 $m^3$ $s^{-1}$) were comparable to the combined stream inflows (0.050 $m^3$ $s^{-1}$). Heat flux analyses indicated that the net advection for the streams and springs were also comparable to each other but were two orders of magnitude less than the downwelling shortwave radiation across the lagoon. Although the lagoon-scale thermal effects of groundwater inflows were small compared to atmospheric forcing, spring discharge dominated heat transfer at a local scale, creating pronounced cold-water plumes along the shoreline.

A numerical model was used to investigate seasonal and multi-decadal groundwater temperature patterns to relate measured spring temperatures to their respective aquifer source depths, and to consider long-term groundwater warming. Based on the different climate scenarios used for 2020 to 2100 (5-year averaged air temperature increase up to 4.32℃), modelled 5-year averaged subsurface temperatures increased 0.08 to 2.23℃ in shallow groundwater (4.2 m depth) and 0.32 to 1.42℃ in the deeper portion of the aquifer (13.9 m), indicating the depth-dependency of warming. This study presents the first analysis of the thermal sensitivity of groundwater-dependent coastal ecosystems to climate change and indicates that coastal ecosystem management should consider the potential impacts of groundwater warming.


## 1 Introduction

Global freshwater temperatures have been increasing in response to changes to climate and landcover (Desbruyères et al., 2017; IPCC, 2014; Isaak et al., 2017; Liu et al., 2020). Water temperature is a critical consideration in aquatic ecophysiology, as it influences the metabolic functions of all organisms (e.g., Morash et al., 2021) and the biogeochemistry of aquatic systems (Ouellet et al., 2020). Cold-water patches, sourced by discrete groundwater inflows to streams, form thermal refuges that enable heat-sensitive species to survive periods of elevated thermal stress (Kurylyk et al., 2015a; Sullivan et al., 2021;

Torgersen et al., 2012; Wilbur et al., 2020). This cooling mechanism depends on the seasonal stability of groundwater temperature relative to surface water due to the insulative effect of ground overlying the source groundwater (Bonan, 2008). In addition to stable temperatures, focused groundwater discharge points in surface water bodies are often characterised by distinct biogeochemical conditions preferred by certain aquatic species (Cantonati et al., 2020; Hayashi & Rosenberry, 2002). Although groundwater-dependent ecosystems may be more resilient to seasonal and short-term extreme weather changes, they

remain susceptible to multi-decadal warming signals that can penetrate deeper into the subsurface to affect groundwater temperatures (Bense & Kurylyk, 2017; Gunawardhana & Kazama, 2011; Menberg et al., 2014).

Surface water temperatures in inland lotic systems are influenced by latent, sensible, and radiative heat fluxes at the water surface, longitudinal heat flux along the channel due to advection and dispersion, and bed heat fluxes due to friction,

conduction, and advection (Caissie, 2006; Dugdale et al., 2017), which in turn are controlled by landscape characteristics (O'Sullivan et al., 2019). The thermal regimes of many coastal aquatic systems are inherently more complex than freshwater systems as they are additionally influenced by exchanges with the ocean (e.g., Newton & Mudge, 2003). Furthermore, vertical and horizontal thermal stratification within coastal waters may arise due to salinity-induced density differences (e.g., Danielescu et al., 2009; Newton & Mudge, 2003; Nunes & Lennon, 1987). Nevertheless, net solar radiation, latent heat of

evaporation, and sensible heat transfer to the atmosphere typically remain the primary thermal drivers in shallow coastal waters (e.g., Ji, 2017; Rodríguez-Rodríguez & Moreno-Ostos, 2006).

Despite the large body of recent work and associated reviews characterising river (Caissie, 2006; Dugdale et al., 2017; Ouellet et al., 2020), ocean (Abraham et al., 2013), and subsurface thermal regimes (Kurylyk et al., 2014a), relatively little work has

focused on the influence of groundwater on the temperature of transitional coastal waters (e.g., Chikita et al., 2015; Rodríguez-Rodríguez & Moreno-Ostos, 2006). Groundwater may be delivered to the coast via direct (e.g., springs) and indirect (i.e., baseflow in streams or rivers) pathways and can influence coastal ecosystems (Luijendijk et al., 2020). As for rivers, groundwater inputs to coastal environments may generate spatial thermal heterogeneity in the receiving water body (e.g., Danielescu et al., 2009; KarisAllen & Kurylyk, 2021), but the ability of these cold-water plumes to serve as thermal refuges is

less explored. Further, although some riverine studies have considered the sensitivity of incoming groundwater to future climate change (e.g., Hannah & Garner, 2015; Kaandorp et al., 2019; Kurylyk et al., 2014b), to our knowledge no studies have

investigated the thermal sensitivity of coastal groundwater discharge to climate change or the potential ecological consequences.

Thermal imaging devices attached to aircraft have been used to aerially map thermal heterogeneity in coastal zones resulting from direct groundwater input (e.g., Coluccio et al., 2020; Danielescu et al., 2009; Lee et al., 2016a). Previous studies have utilized thermal infrared imagery to estimate local groundwater discharge via empirical relationships with thermal plume geometry (e.g., Bejannin et al., 2017; Danielescu et al., 2009; Kang et al., 2019; Kelly et al., 2019b; Lee et al., 2016a; Mundy et al., 2017; Tamborski et al., 2015). Small rotary-wing drones have the capacity to inexpensively collect thermal data with

higher temporal and spatial resolution relative to conventional occupied aircraft (Dugdale et al., 2022; Lee et al., 2016b), although drone thermal data often involve additional challenges (e.g., thermal drift and limited spatial coverage; Dugdale et al., 2019; Kelly et al., 2019a). Despite these issues, this technology is suitable for determining relative temperature differences in individual images and thus can be used to locate focused groundwater inputs that generate anomalous water temperatures.

The overall goals of this study were to (1) quantify the discharge and present thermal influence of inter-tidal springs in a warm coastal lagoon ecosystem and (2) investigate how these springs will be thermally impacted by climate change. Drone thermal imaging was paired with *in-situ* thermal and hydrologic monitoring to locate and further investigate spring and groundwater-dominated stream inputs to the lagoon. Comparison to stream inputs was conducted to emphasize the relative importance of focused intertidal groundwater discharge at this site. Spring discharge estimated via drone thermal imagery was compared with

total direct groundwater input estimated by using radon as a groundwater tracer. To interpret our measured spring temperatures and better understand how springs will respond to future warming, a numerical heat transfer model was applied to relate measured seasonal temperature signals at springs to their respective aquifer source depths and to simulate depth-dependent aquifer warming due to climate change between 2020 and 2100.

## 2 Site description

The study took place in the Basin Head lagoon on the eastern shore of Prince Edward Island (PEI) in Atlantic Canada (Fig. 1). The lagoon was established as a Marine Protected Area in 2005 under the *Oceans Act* to protect giant Irish moss, a unique morphotype of Irish moss (*Chondrus crispus*) endemic to the lagoon (DFO, 2009). The biomass of giant Irish moss within the lagoon declined by over 99% from 1980 to 2008 (DFO, 2009), and thermal stress has been identified as one of the compounding stressors contributing to its decline (Joseph et al., 2021). The Basin Head lagoon is approximately 0.6 km$^2$, with water depths

that rarely exceed 2 m at high tide. The lagoon has a mixed semi-diurnal tide, with an average range of approximately 0.8 m, and is connected to the ocean by a narrow, artificial channel (Fig. 1b).





PEI is characterized by mean annual precipitation ranging from 1046 to 1241 mm yr$^{-1}$ and mean monthly air temperatures from -7.9 to 18.6°C based on historical records of eight Environment and Climate Change Canada (ECCC) weather stations

(Rivera, 2014). Precipitation is routed from the Basin Head watershed to the lagoon via groundwater-dominated streams (Fig. 1b) and direct groundwater discharge pathways. PEI bedrock aquifers are typically weakly consolidated, very fine to coarse, fractured sandstones with sparse occurrences of mudstone, conglomerate, and/or breccia (Brandon, 1966; Crowl, 1969a; van de Poll, 1989). Surficial tills within the study watershed are mainly clay-sand to sand phase tills (Crowl, 1969b; Prest, 1973) and are estimated to be 5 m deep on average based on local core logs (Government of PEI, 2019).

**3 Methods**

Field work and data collection, including the instrumentation of springs and streams and the installation of a climate station and coastal piezometer (Fig. 1b), for this study occurred between June 2019 and November 2020. Lagoon water temperatures typically peak in July and August in the Basin Head lagoon, which reflects the period of greatest thermal stress for giant Irish moss. Contrast between groundwater and lagoon water temperatures is also greatest in July and August, which is favourable

for the detection of springs via thermal infrared imaging. Accordingly, a dense network of sensors (Fig. 1) was temporarily installed between July 23 and August 26, 2020, to provide a more detailed assessment of groundwater discharge (i.e., the 35-day 'focused study period') during this critical period.  Also, drone thermal images were captured in the summer of 2020, and radon sampling occurred during the summer and fall of 2020.

**3.1 Remote thermal sensing and relationship to spring discharge**

Stationary nadir thermal infrared images were taken (within ±2 hours of low tide, from an elevation of approximately 60 masl, during clear sunny days) of the springs entering the lagoon throughout July and August 2020. This study used a Matrice 210 RTK v2 aerial drone, equipped with a 13 mm non-radiometric DJI ZENMUSE™ XT2 thermal infrared camera with FLIR technologies (XT2; DJI, 2018). Real-time kinetic processing was used for drone navigation, as well as image geotagging, and the position of the images relative to the base station is expected to be highly accurate (<5 cm) even without the use of ground

control points (Kalacska et al., 2020). The XT2 has a 45°×37° field of view, 640×512 resolution, 8-bit colour pallet, spectral range between 7.5 and 13.5 μm, sensor sensitive range between -25 and 135°C (*High Gain Mode*), and an absolute thermal accuracy of ±5 to 10°C (DJI, 2018). The absolute temperature of the thermal imagery was not deemed reliable due to internal drift of the sensor, lack of radiometric correction, and disagreement in thermal readings between frames. However, it was assumed that the relative temperature data in each frame were sufficiently precise for the consistent definition of thermal plume

geometry, given the reproduceable ability of the XT2 to identify surficial thermal anomalies confirmed with *in-situ* temperature measurements. Rather than developing a per-pixel corrections matrix for the sensor to correct for distortion towards the image periphery, only the central portion of each image was analysed (Kelly et al., 2019a).



This study applied FLIR Tools®, ImageJ, and MATLAB® to post-process grayscale intensity data from the thermal infrared
images using the procedure summarized in Fig. 2. These products enabled the analysis of high-resolution thermal data and
polygonal cropping procedures. Grayscale intensity data was extracted from the thermal images of the spring-sourced plumes
and graphed with respect to cumulative area to yield a characteristic S-shape type-curve (Fig. S1). Each 'inflection point' of
the graph was used to define 'thermal groups' and the sharp transition zones between them (Roseen, 2002). An empirical
relationship was developed between discharge measurements for a subset of springs (Sect. 3.2) and the area of spring thermal
plumes determined from the graphical analysis (e.g., Danielescu et al., 2009). This plume size-spring discharge relationship
was then applied to estimate the *instantaneous* discharge of ungauged springs from their respective thermal plume areas
captured by drone thermal imagery. *Continuous* spring discharge to the lagoon was estimated for the focused study period
using a hydrologic proxy (e.g., Danielescu et al., 2009). Herein, the water levels in our near-shore piezometer (Fig. 1b and
Section 3.2) were used as a proxy for the aquifer-lagoon hydraulic gradient and spring discharge via proportionality constants
developed from the drone-based instantaneous discharge estimates (i.e., discharge was assumed to vary linearly with
piezometer water table). Approximately 20% of the lagoon's north-western shoreline could not be surveyed with the drone
based on proximity to the road or power lines, but the presence of springs along this unsurveyed portion has been confirmed
by distant thermal images and *in-situ* measurements. Consequently, the total spring discharge to the lagoon was estimated by
extrapolating the average spring discharge per shoreline length obtained from the surveyed segments (80%) to the unsurveyed
segment (20%).

## 3.2 Hydroclimatic, thermal, and radon monitoring

The manufacturer, model, location, and monitoring durations for each logger are listed in Table S1, and locations are visually
portrayed in Fig. 1. A climate station (Onset Hobo Micro Station Logger) was installed at the study site to measure
downwelling shortwave radiation, windspeed, rainfall and air temperature. Also, subsurface modelling and hydraulic
assessments were guided by *in-situ* field measurements of subsurface properties using a shallow groundwater piezometer (5
masl; Fig. 1b). The piezometer fully penetrated the surficial soils to a depth of 4.5 m. This lowland well was instrumented with
a pressure transducer to monitor well recovery during a slug test, as well as to provide a record of water table dynamics,
groundwater temperature, and electrical conductivity. Water stage was monitored at 15-minute intervals in the four primary
streams (S1 to S4; Fig. 1b) over the study period using pressure transducers corrected with air pressure data from the nearest
ECCC climate stations (Station IDs 41903 and 7177; ECCC, 2021a, 2021b). Stream discharges were measured via the velocity-
area (Dingman, 2002, p. 609) method using a Sontek FlowTraker2® (Xylem Inc, Rye Brook, New York, USA) and were used
to generate rating curves for local streams (average n = 6 and $R^2$ = 0.937). Other smaller streams (S5 and S6, Fig. 1b) were
gauged intermittently, but their flow rates were < 1% of the combined flow of streams S1 to S4 and are thus hereafter not
considered. Considering the limited amount of precipitation (35.6 mm) over the 35-day focused study period (July 23 to August
26, 2020), streamflows were assumed to be entirely baseflow. This simplification will be assessed later but is not anticipated



to introduce significant error because PEI streams have frequently been documented to be 80-100% baseflow during the summer (Benson et al., 2007; Brandon, 1966).

To develop relationships between spring discharge and thermal plume area (Sect. 3.1), volumetric flow measurements for three springs (Figs. S2, S4, yellow rings) were conducted at low tide by constructing custom weirs surrounding their respective outlets. Three springs were chosen to represent the range of anticipated spring discharges to the lagoon. Volumetric measurements of springs were made with an accuracy of ±10 mL, but flows were not entirely captured at the spring outlet due to limitations with the water collection technique and outlet geometry. To remove any tide-circulated saltwater from our spring

discharge estimates (LeRoux et al., 2021), the freshwater component discharging from the spring was isolated by estimating saltwater content via a simple two-component mixing model based on the electrical conductivities of the spring, lagoon, and shallow fresh groundwater.

Additional instruments were installed throughout the lagoon and watershed (Fig. 1b) in tandem with stream monitoring work

to investigate water quality and hydrologic/hydrodynamic processes. Temperature sensors were installed at multiple locations along the lagoon channel at the top (affixed to a buoy) and bottom (affixed to an anchor) of the water column, three springs outlets (i.e., Springs 2, 5, and 21; Figs. S2 to S5, blue rings), and the four primary streams (Streams S1-S4) to characterize their thermal regimes. Spring temperatures patterns (i.e., seasonal amplitudes) were compared to the results of the thermal numerical modelling (Sect. 3.3) to estimate the aquifer source depth for a given spring following the effective aquifer depth

approach of Kurylyk et al. (2015b) and Briggs et al. (2018). Also, the paired spring flow and temperature data were used to quantify net (sensible) advective heat fluxes to the lagoon over the focused study period (Kurylyk et al., 2016):

$$J_{adv,i} = C_w Q_{input}(T_{input} - T_{lagoon}),\qquad\qquad\qquad(1)$$

where $J_{adv}$ is the net (sensible) advective water energy flux (W), $C_w$ is the volumetric heat capacity of water (J m$^{-3}$ °C$^{-1}$), $Q_{input}$ is the input (direct rainfall, spring, or stream) water discharge (m$^3$ s$^{-1}$), $T$ is the water temperature (°C), $T_{input}$ and $T_{lagoon}$ are the water temperatures for the hydrologic input (rainfall, spring, or stream) and lagoon, respectively. Precipitation temperature was assumed to be the same as the average air temperature from the climate station over the short, focused study period.


Advective heat fluxes for the springs and streams were considered to integrate the hydrology and thermal investigations in this study and to investigate the springs' thermal function in the lagoon. A complete lagoon energy balance cannot be completed due to a lack of complete surface energy flux data and data for the hydraulic and thermal exchange with the ocean. However, as a first-order estimate of the relative thermal effects of the freshwater inflows at the lagoon scale, the advective fluxes



obtained via Eq. (1) were compared to the downwelling shortwave radiation (W m$^{-2}$) measured at the study site climate station
and multiplied across the lagoon surface area.

An electrical conductivity logger was installed in the largest stream (S1). Conductivity-temperature-depth loggers (Solinst
Levelogger® 5 LTC) were installed within the lagoon and in two intertidal springs (summer 2020 only). Discrete water
temperature and electrical conductivity measurements of the lagoon, springs, streams, and piezometer were also taken during
field investigations using handheld devices (Apera EC400S Portable Conductivity/TDS/Salinity/Resistivity Meter and a YSI
ProDSS Multiparameter Digital Water Quality Meter) and a Solinst LTC logger to parameterize the two-component salinity
mixing model.

Dissolved radon ($^{222}$Rn; $t_{1/2}$ = 3.83 d) is naturally enriched in groundwater and is an inert noble gas, making it an effective
tracer for groundwater discharge to coastal systems (Swarzenski, 2007). Four groundwater springs were sampled for $^{222}$Rn in
August and November 2020 (Fig. 1b) coincident with continuous paired electrical conductivity, water depth, and temperature
monitoring as previously described. Glass bottles (250 mL) were submerged directly at the spring outlet and allowed to
overflow, collected bubble-free without headspace, and analysed via RAD-H2O (Durridge Co.). Stream surface waters and
shallow lagoon pore waters were additionally analysed in November (Fig. 1b). Near the inlet of Basin Head lagoon, surface
water was continuously drawn into a gas exchange chamber (RAD-AQUA), and $^{222}$Rn was monitored using a commercial
radon-in-air monitor (RAD7, Durridge Co.) over 24 hours in August (Fig. 1b, southernmost blue ring). Dissolved $^{222}$Rn
activities were determined using the solubility constants from Schubert et al. (2012) for temperature and salinity and corrected
for instrument response delay.

A mass balance model was developed for $^{222}$Rn (Burnett & Dulaiova, 2003; Rodellas et al., 2021; Sadat-Noori et al., 2015):

$$J_{mix} + J_{decay} + J_{atm} = J_{spring} + J_{stream} + J_{diff} + J_{Ra-226},$$  (2)

where $J$ represents the flux of $^{222}$Rn (Bq d$^{-1}$) for all known sources (baseflow-fed streams; molecular diffusion; $^{226}$Ra
production) and sinks (mixing; radioactive decay; atmospheric evasion) of $^{222}$Rn within the Basin Head lagoon. With the time-
series monitoring station near the inlet of the lagoon, we assume that this point-in-space is representative of all $^{222}$Rn inputs
and outputs through the tidal inlet and thus any imbalance between known sources and sinks is attributed to unknown
groundwater inputs ($J_{spring}$). This estimate provides a maximum range of groundwater inputs (Peterson et al., 2010).

**3.3 Groundwater and thermal numerical modelling**

A 1-D subsurface heat and water transport model was developed and manually calibrated to local groundwater temperature
observations, with hydrologic parameterization informed by local data (e.g., weather data, piezometer slug test) and literature



values. Downscaled future climate projections were then applied as upper boundary conditions to drive simulations of plausible future subsurface temperatures, with the goal of assessing the potential sensitivity of springs to projected multidecadal warming

trends (Fig. 3a). The conceptual complexity of the numerical model was limited both to facilitate model parameterization as well as interpretation; nevertheless, this approach preserved key heat transport processes. Multi-dimensional systems such as the fractured sandstone/mudstone aquifers feeding the intertidal springs in the Basin Head lagoon may be simplified into a one-dimensional system operating on the concept of an 'effective aquifer depth', which lumps multi-dimensional processes and can be derived by relating the amplitude decay or phase shift of the seasonal groundwater temperature sinusoid relative to

the air temperature signal (Kurylyk et al., 2015b). One-dimensional heat transfer modeling approaches have been used in previous studies considering groundwater thermal impacts on rivers (e.g., Briggs et al., 2018a, b) and in analytical solution studies of past or future groundwater warming (e.g., Gunawardhana et al., 2011; Irvine et al., 2017). The thermal regimes of shallow aquifers exhibit a depth-dependent response to seasonal surface temperature signals and climate change, and thus the measured seasonal amplitude of groundwater discharge temperature yields an approximate average groundwater depth

(Kurylyk et al., 2015b) that can be used to estimate the thermal response of that spring to multi-decadal warming.

The selected model, Simultaneous Heat and Water model (SHAW; Flerchinger & Saxton, 1989), simulates transient vertical energy and water transport through a canopy, snow layer, plant residue, and soil layers (Flerchinger, 2017). The robust physical basis and ability of SHAW to simulate the surface energy balance, snowpack, vegetation, and seasonally frozen soil processes

(e.g., Mohammed et al., 2017) made it an appealing choice for this long-term thermal study, as these processes affect subsurface thermal trends at the latitude of the study site. A detailed description of model processes and equations, as well as the boundary condition options, are detailed in Flerchinger (2017). Standard values were employed for the thermodynamic properties of water (Flerchinger and Saxton, 1989). Bulk thermal properties of the subsurface in SHAW are estimated based on the approach of DeVries (1963) by using user-input soil compositions and model-computed water content; soil compositions were herein

based on local soil surveys and historical studies of PEI soils (e.g., Crowl, 1969a). This study separated the model domain into an unsaturated upper region (0 to 3 m depth) that computed the upper boundary condition and forcing to the lower, saturated region model (3 to 93 m depth; Fig. 3b). SHAW version 3.0.3 was used for the upper domain to calculate surface and vadose zone processes, whereas a modified version of SHAW 2.4 was used for the lower region to exclusively consider subsurface thermal transport below the water table without solving the surface energy balance (Mohammed et al., 2017).


Climate inputs required by SHAW to solve the surface energy balance for the upper region model include maximum and minimum daily air temperature, dew point temperature, wind speed, total precipitation, and all-sky radiation. The timestep, input data, and output of the simulations had a daily resolution. Based on the period of this study and the availability of historic data and climate projections, historical simulations were conducted over 37 years (1984-2020), and future simulations were

run over 81 years (2020-2100). The minimum and maximum air temperatures, as well as total precipitation for the historical simulations, were sourced from the CNRM-CM5, RCP4.5 hindcast model (Voldoire et al., 2013), which more accurately





reproduced historical conditions for PEI locations relative to other climate simulations (Warner, 2016). Local dew point temperature, wind speed at 2 m above ground level, and all-sky solar radiation data were sourced from the NASA POWER reanalysis database (Sparks, 2018). As there were no readily accessible future projections for dew point temperature, wind
speed, and all-sky solar radiation, these were estimated by repeating data from a portion of the historical period (i.e., 1985-2020; Sparks, 2018). The repeating of these data is not expected to produce significant errors given the relative hydraulic and thermal inertia of groundwater systems and because groundwater temperature changes are later interpreted herein using 5-year averages to smooth out any short-term effects. Future daily maximum air temperature, minimum air temperature, and total precipitation were sourced from four climate simulations based on work by Warner (2016): (1) CNRM-CM5, RCP4.5; (2)
CNRM-CM5, RCP8.5; (3) MRI-CGCM3, RCP4.5; and (4) MRI-CGCM3, RCP8.5 (ECCC et al., 2021). Simulated temperature at 3 m in the upper region model was then used as the upper boundary condition for the lower saturated model (Fig. 3b).

## 4 Results

### 4.1 Remote thermal sensing and spring discharge analysis

Based on cold-water plumes visible in the drone-based aerial thermal imagery (see Fig. 4 for examples), 40 springs were
located on the north and west shores of the lagoon. These are mapped in Fig. 1b, with enhanced zoom and labels in Figs. S2-S5. Selected springs identified from the thermal imagery were gauged (Table S2) to develop a plume size-discharge relationship (Fig. 5d and Sect. 3.1-3.2). Electrical conductivity values for the low-low tide discharge measurements of the three gauged springs and the associated end-member analysis revealed that spring discharges at the times of measurement were <2% saltwater, so the resultant freshwater correction had a minimal effect on discharge estimates. The paired discharge values and
thermal plume areas for the three gauged springs yielded a power function relationship for the lagoon ($R^2$=0.99; Fig. 5d).

The area of only 34 springs were graphically assessed using low-low-tide thermal image pixel data (Table S2) because the remaining identified springs were either too small or inaccessible via drone. The workflow and resulting plume area associated with Spring 8 is shown as an illustrative example in Fig. 5. Instantaneous spring discharges for ungauged springs (Springs 1-
31; Table S2) were computed as a function of plume area using the lagoon power function (Fig. 5d). Only Spring 1 had a larger plume size than the largest gauged spring (Table S2), indicating that the discharge values for the springs were generally constrained by the area range in our empirical plume area-discharge relationship. The estimation of continuous spring discharge over the focused study period from the instantaneous spring discharges via continuously available proxy data (i.e., piezometer water level) yielded a total spring discharge volume estimate for this 35-day period of 113,000 $m^3$ (0.037 $m^3$ $s^{-1}$).
Springs were found at a density of approximately six springs per kilometre along the surveyed section, which yielded an estimate of approximately 580 $m^3$/km/day (0.0067 $m^3$/s/km) for the discharge rate per shoreline length. Assuming a constant spring flow density for the 20% unsurveyed shoreline resulted in a cumulative estimated 35-day total spring discharge of 142,000 $m^3$ (0.047 $m^3$ $s^{-1}$).



## 4.2 Hydroclimatic monitoring data and analyses

### 4.2.1 Stream discharge monitoring results

Stream monitoring data (Fig. S7) were analysed to estimate the total indirect groundwater flow (baseflow) to the lagoon during the focused study period, which yielded the following inflow volumes (flows): S1 = 90,000 m$^3$ (0.030 m$^3$ s$^{-1}$); S2 = 22,000 (0.0073); S3 = 33,000 (0.011); and S4 = 7,700 (0.0025). Based on the assumption that all streamflow is baseflow during the summer months as supported by the lack of flow 'spikes' (Fig. S7) and typical summer conditions in PEI, streams contributed approximately 153,000 m$^3$ (0.050 m$^3$ s$^{-1}$) of indirect groundwater to the lagoon over the focused study period. This total streamflow is within 6% of the total spring inflow estimated from the thermal analysis, suggesting the two hydrologic pathways for groundwater delivery (baseflow and spring discharge) are comparable at this site in the summer.

### 4.2.2 *In situ* temperature data

Water temperatures in the lagoon were high during the focused study period (maximum temperature of 33°C), with mean daily water temperatures often greater than the mean daily air temperatures and occasionally exceeding 25°C in the northeast arm of the lagoon (Fig. 6). In contrast, the groundwater-dominated streams had mean daily water temperatures between 10°C and 14°C during this period, and groundwater discharge temperatures remained between 7 and 10°C for all continuously monitored springs (Fig. 6). Seasonal lagoon water temperatures peaked in late July to early August. Lagoon and stream temperatures exhibited at least limited diel variability (hourly data, Fig. S8), whereas none of the monitored springs displayed diel temperature trends once tidal effects were removed. Over the focused period, the median 15-minute water temperatures and interquartile ranges (IQR) of Stream S1, S2, S3 and S4 were 8.7°C (IQR = 0.6°C), 10.8°C (IQR = 1.2°C), 10.5°C (IQR = 1.3°C), and 10.4°C (IQR = 1.0°C), respectively. Stream temperature measurements were taken near the stream mouths (above normal head of tide) and represent the outcome of the cumulative upstream heat exchange, including the surface heat fluxes absorbed along the channel that contributed to the stream temperatures exceeding the spring temperatures in the summer months (Figs. 6 and S8). Five temperature sensors distributed throughout the lagoon (Fig. 1b) over the focused period yielded a higher temperature median (~22°C) and variability (IQR = 4°C). Temperatures were typically greatest in the shallower, more poorly flushed upper reaches of the northeast arm of the lagoon and lowest in the deeper main basin (Figs. 1b, 6).

Summertime lagoon water temperatures over the study period were consistently lowered surrounding spring outlets, enabling the drone-based analysis in this study; however, the extent of these thermal anomalies varied substantially with tidal stage and channel geometry (KarisAllen & Kurylyk, 2021). The difference between coincident spring and lagoon temperatures was up to 23°C (Figure S8b). The thermal patterns of three springs (Fig. 7) were analysed to estimate their seasonal signal properties (especially amplitude) and by extension their relative depth and vulnerability to climate warming. Temperatures at each of the spring outlets (Fig. 7) exhibited pronounced semi-diurnal oscillations (i.e., 12.42 hr periods) due to the altered aquifer-lagoon hydraulic gradients and enhanced lagoon mixing at higher tide. The stability of the actual groundwater discharge temperature



over tidal periods was confirmed by one sensor buried slightly deeper (5-10 cm) in Spring 3 that only exhibited seasonal variation (not shown). To isolate the groundwater temperature from the time series at the spring outlets, the temperatures at low tide over several months of tidal cycles were fitted with an annual (period = 1 year) thermal sinusoid (red dashed lines, Fig. 7). The average temperature of Spring 5 was 7.65°C (Fig. 7a). The lack of thermal periodicity in this spring suggests that

its source depth is below the extinction depth of annual air temperature patterns (normally 10-20 m in this region, e.g., Kurylyk et al., 2015b). In contrast, Spring 21 (Fig. 7b) displayed an annual signal with a mean of 7.75°C and an amplitude (half the range) of 1.6°C. Spring 2 also displayed a seasonal signal (Fig. 7c) with the lowest mean (7.05°C) and the highest amplitude (2.0°C). This amplitude suggests that Spring 2 has the shallowest source depth and is the most vulnerable to multidecadal warming of the three springs investigated as discussed later. The fitted spring annual temperature amplitudes were later

compared to depth-variable seasonal results from numerical modelling to infer approximate average depths of the groundwater delivered to the springs.

### 4.2.3 Lagoon heat fluxes

Selected advective components of the Basin Head lagoon heat budget associated with freshwater inflows were estimated for the 35-day focused study period (Table 1). Continuous spring discharge for the net advection calculation was estimated from

the water table proxy approach (Sect. 3.2). The freshwater inflows from the precipitation, streams, and springs cooled the lagoon water temperature over the summer, as indicated by their negative net thermal advection values (Eq. 1) in Table 1. The estimated total net advective heat flows for the streams and springs were almost identical and over an order of magnitude higher than the advection from direct precipitation. Any unquantified diffuse groundwater input (upwelling to lagoon) would further increase the relative contribution of direct groundwater on the lagoon heat budget. As expected, heat flow from

downwelling solar radiation was substantially larger than advective heat components to the lagoon (Table 1), suggesting that the springs and streams likely exert minor influence on the average water temperatures throughout the lagoon, despite their evident thermal impact at a localised scale along the shoreline (Figs. 4 and S8). A heat budget, including advective exchanges with the ocean and a complete surface energy balance, is required to gain a full understanding of the relative thermal effects of these freshwater inflows at the scale of the full lagoon, but data are not available for many heat flux components.

### 350 4.2.4 Radon results

Near the lagoon inlet, surface water $^{222}$Rn activity varied from 10 to 97 Bq m$^{-3}$, with maximum activities occurring near low tide when salinities were lowest, and following classic hysteresis loops (Figs. 8a, b). The $^{222}$Rn activity of the fractured sandstone springs (10,400 ± 3,700 Bq m$^{-3}$; n=4) were an order of magnitude higher than shallow, brackish porewaters (630 ± 250 Bq m$^{-3}$; n=4) and baseflow-fed streams (1,100 ± 1,200 Bq m$^{-3}$; n=4) as shown in Fig. 8a and Table S3. Stream discharge

during the surveyed period, 0.05 m$^3$ s$^{-1}$, results in a stream-derived radon flux of (4.7 ± 5.6) × 10$^6$ Bq d$^{-1}$. This flux represents a theoretical maximum, as there will be appreciable $^{222}$Rn degassing and decay within the stream prior to entering the lagoon. Based on the minimum observed $^{222}$Rn concentration (Gilfedder et al., 2015), the diffusive flux of $^{222}$Rn may be approximated





as 11 ± 6 Bq m$^{-2}$ d$^{-1}$; or (6.4 ± 3.2) × 10$^6$ Bq d$^{-1}$, over the total lagoon area. Losses of $^{222}$Rn due to tidal mixing (Burnett &
Dulaiova, 2003) and atmospheric evasion (MacIntyre et al., 1995) are taken as the mean (± standard deviation) losses estimated
over the 24-hour tidal cycle, upscaled to the lagoon surface area (Table S4). Similarly, radioactive decay is estimated
considering the mean excess $^{222}$Rn inventory, for a net loss of (1.9 ± 1.6) × 10$^6$ Bq d$^{-1}$. Considering known sources and sinks,
there is an excess of $^{222}$Rn (8.0 ± 6.0 × 10$^7$ Bq d$^{-1}$) attributable to groundwater. Using a $^{222}$Rn endmember from the fractured-
sandstone springs (10,400 ± 3,700 Bq m$^{-3}$), we estimate maximum groundwater inputs of 0.09 ± 0.07 m$^3$ s$^{-1}$. Given our
uncertainties, the absolute value of this flux should be interpreted with caution, but it is useful for placing results from other
methods into a broader context.

## 4.3 Groundwater and thermal numerical modelling results

### 4.3.1 Model calibration and sensitivity

Model parameters were manually calibrated to improve agreement of the historical simulation with the approximate calibration
targets. A fixed water table depth of 3 m relative to ground surface was assumed based on this piezometer's monitoring data
over the study period (June 2019 to November 2020). The SHAW model was manually calibrated to the mean subsurface
temperatures measured in this piezometer, as well as the amplitude attenuation and lag of the annual groundwater temperature
signal relative to the air temperature signal. The piezometer sensor was at a depth of 4.24 m below surface and recorded
groundwater temperatures between 5.10 and 9.50℃, annual amplitudes between 1.80 and 2.20℃, and a lag of 70 to 100 days
relative to the annual air temperature signal based on 2019 and 2020 data. The outputs of the calibrated historical simulation
were in reasonable agreement with the piezometer data. The range of mean annual temperatures, as well as the amplitude and
lag of the thermal signal at each depth were calculated using the final 5 years of the historical simulation (i.e., 2016-2020). At
4.2 m depth, the modelled 2016-2020 mean annual groundwater temperature was between 7.45 and 7.8℃, the amplitude was
2.1 to 2.2℃, and the lag was 92-105 days. Furthermore, after accounting for the difference in water table depth, modelled
outputs at 13.9 meters depth were in agreement with temperature measurements at the same depth in a nearby upland provincial
observation well (55 masl; Government of PEI, 2021).

### 4.3.2 Historic and future simulation results

The final 5 years of the future simulations (2096-2100) were compiled and compared to the final 5 years of the historical
simulation (2016-2020, Table 2). The subsurface temperatures at 4.2 and 13.9 m (piezometer and government well sensor
depths) increased with increasing atmospheric and surface temperatures in all simulations (Fig. 9). Modelled groundwater
temperature is projected to increase by 0.08 to 2.23℃ at 4.2 m and 0.32 to 1.42℃ at 13.9 m, indicating the depth-dependency
of warming for a given timeframe and the influence of a given climate scenario. The MRI-CGCM3, RCP 8.5 simulation had
the greatest temperature increase, and the MRI-CGCM3, RCP 4.5 simulation had the lowest (Table 2).



The atmospheric forcing (Fig. 10a) and subsurface temperature response (Fig. 10b) over the last five years (2016-2020) of the historical simulation are presented for different depths to illustrate the intra-annual variability of temperature and the attenuation and lagging of the surface temperature signal with depth. The modeled amplitudes of the annual temperature signals (Fig. 10) may be compared to the measured spring outlet thermal patterns (red lines, Fig. 7) to estimate the springs' effective source depths (Kurylyk et al. 2015b). Based on their annual amplitudes, Springs 2 and 21 are likely sourced from depths

between 3 and 7 m, whereas Spring 5 is interpreted to be predominantly fed from depths below 12 m.

The notion of diverse (i.e., depth-dependent) spring thermal sensitivities is further supported by comparing the warming rates at different depths within the soil column. For example, 5-year averaged air temperature is simulated to increase by approximately 4.32°C over the course of the warmest future simulation (i.e., MRI-CGCM3, RCP8.5). This air temperature

signal increased the 5-year averaged groundwater temperature by approximately 1.78°C at 4.2 m depth and 1.57°C at 13.9 m depth. For relative comparison, this suggests a relative warming rate of 0.41°C at 4.2 m depth and 0.36°C at 13.9 m depth per 1°C of air temperature rise by the year 2100. The model results also illustrate that shallower aquifer zones are more vulnerable to short-term (seasonal and inter-annual) variations in temperature given how the seasonal amplitude and year-to-year variation are reduced with depth (see Fig. 9a,b and 10b).

**5 Discussion**

**5.1 Thermal plume analysis and continuous discharge estimation**

This study applied a power curve regression to the collected spring discharge and area data, which varies from previous studies that have applied linear (e.g., Bejannin et al., 2017; Lee et al., 2016b; Tamborski et al., 2015) or logarithmic relationships (Danielescu et al., 2009). Our high coefficient of determination ($R^2 = 0.99$, Fig. 5d) suggest a strong relationship between

plume size and discharge, although we concede this is based on limited points. Also, previous studies have converted instantaneous discharge measurements based on thermal plume analysis to continuous discharge estimates by using baseflow as a proxy for spring discharge (Bartlett, 2011; Danielescu et al., 2009). Rather than baseflow, we used groundwater levels measured in a piezometer relatively close to the lagoon as this was thought to be a better proxy for the local hydraulic gradient (and thus spring flow) than baseflow which integrates processes further up-catchment.


To overcome limitations with the number of points informing the thermal plume area-discharge relationship and the associated total spring discharge estimate of 0.047 m$^3$ s$^{-1}$, we independently assessed total groundwater inputs using a $^{222}$Rn mass balance. Assuming that groundwater discharge to the lagoon accounted for the differences between known $^{222}$Rn sources and sinks, maximum input of groundwater was estimated as 0.09 ± 0.07 m$^3$ s$^{-1}$ (Table S4). Given the uncertainty of both approaches,

these independent assessments are quite comparable. Also, the $^{222}$Rn approach may capture additional diffuse groundwater



inflows not captured by the drone survey, and thus it is expected the discharge from the radon approach would be higher. For example, Danielescu et al. (2009) found that approximately 25% of groundwater inflow to two PEI coastal systems was diffusive, and such inflows were not accounted for in the drone thermal imagery analysis in this study. The results reveal the value in using complementary but independent estimates of groundwater inflows from different types of tracers (herein heat and radon), particularly if both estimates are highly uncertain.

The comparison of estimated streams and spring flows from this study reveal that the magnitude of direct groundwater inputs to PEI coastal systems is likely significant relative to stream inputs. As in other studies (Danielescu et al., 2009), we assumed that intertidal spring discharge measurements taken at low tide were representative of the discharge over the tidal cycle. However, discharge would theoretically decrease at higher stage due to the reduced aquifer-lagoon hydraulic gradient (Lee et al., 2016b; LeRoux et al., 2021), and spring-sourced thermal plumes at this site can be obscured at high tide (KarisAllen & Kurylyk, 2021). This is supported by time-series observations of $^{222}$Rn, where maximum activities are observed during ebb and low tides (Fig. 8c). However, relatively low electrical conductivity and temperature around certain springs during high tides suggests that at least some discharge continuously.

## 5.2 Water temperature and heat transfer

The thermal imagery and the *in-situ* temperature time series reveal the contrast between summer 2020 lagoon temperatures (mean ~ 22°C, maximum 33°C) and the stream (8-13°C) and spring temperatures (7-10°C). The relative hydrologic and thermal stability of the streams attest to their groundwater dominance (Kelleher et al., 2012; Mayer, 2021; Johnson et al., 2021). The *in-situ* data and thermal imagery also collectively illustrate that thermally stable groundwater inflows can reduce the *temporal* variability in surface water temperature (streams vs. lagoon temperatures, Fig. 6) and yet simultaneously enhance the *spatial* variability of temperature (lagoon cold-water patches). The influence of groundwater on the lagoon temperature, relative to other thermal controls (e.g., tidal exchange, solar radiation), is likely dynamic in space and time. Groundwater inputs may be most significant as a thermal buffer throughout the hottest periods of the summer months when rainfall is scarce and lagoon temperatures and stream baseflow indices peak. It is expected that groundwater influence is more impactful overnight, in the absence of solar radiation, and during low tides when spring discharge is potentially at its greatest and the total volume within the lagoon is reduced. A full lagoon energy budget (e.g., Rodríguez-Rodríguez & Moreno-Ostos, 2006) would improve our understanding of lagoon-scale thermal dynamics and thus the larger-scale significance of groundwater and its sensitivity to climate warming. However, at a local scale, cold-water plumes created by inter-tidal springs can create distinct thermal zonation (e.g., Figs. 4, S8) that could potentially provide thermal relief to aquatic organisms capable of behavioural thermoregulation or to static organisms collocated with the discharge point. While such groundwater-sourced, thermally habitable niches have received considerable attention in freshwater environments (Torgersen et al., 2012; Sullivan et al., 2021), they are less studied in transitional, coastal waters (Lecher and Mackey, 2018). The identified cold-water plumes are





concentrated along the shoreline (Fig. 1, grey circles), indicating that the nearshore zone and associated microecosystems may be more strongly influenced by focused groundwater inflows than far-shore zones within this coastal lagoon.


## 5.2 Modelling implications

Intertidal springs in the lagoon are sourced from different effective depths in the groundwater system(s). Individual springs experience varied thermal forcing based on their associated soil layers, land-use, land cover, and travel paths that dictate their thermal signature and sensitivity to surface temperatures. In this study, a one-dimensional subsurface model was used to

demonstrate that springs within the lagoon are expected to warm in response to future atmospheric warming within decades. The reduced groundwater warming compared to atmospheric warming (Sect. 4.3.2 and Fig. 9) does not imply that aquifers ultimately attenuate multi-decadal surface warming signals, but rather that there is a lag between a surface warming signal and its subsurface manifestation (Menberg et al., 2014; Bense and Kurylyk, 2017). For example, if the climate warmed to 2100 and then stabilized, the shallow aquifers would eventually be in equilibrium with the new thermal conditions and the associated

damping of groundwater warming relative to atmospheric warming would become progressively less apparent. It is also important to note that the lag in groundwater warming in response to climate change is not the same as the lag in response to seasonal forcing (Section 4.3.1), because the lag depends on the period of the forcing signal (e.g., Stallman, 1965). Modelling results suggest that the mean annual temperature of shallower groundwater supplying some springs may warm more than 2°C before the year 2100 (Table 2). The overall distribution of spring source depths would need to be further explored (e.g., with

tracers to estimate groundwater residence time) to assess how sensitive groundwater inputs to Basin Head lagoon may be at the lagoon scale, but these modeling results are valuable to understand the present/future system and to inform future research and management initiatives in this Marine Protected Area (see Joseph et al., 2021).

Considering the data availability and modelling objectives, the resulting calibration and model application were considered

satisfactory for the investigations described above. However, future work could consider warming in a multi-dimensional aquifer system with responsive water table dynamics or more fully integrate the lagoon within the model domain in a coupled groundwater-surface water thermal modeling framework (e.g., Brookfield et al., 2009). Numerical groundwater models that account for secondary porosity could be used to consider heat transfer within the fracture network and the porous sandstone matrix (Graf & Therrien, 2007).

## 5.4 Ecological implications of spring warming

Springs are known to support critical groundwater-dependent ecosystems (Cantonati et al., 2020) due to the distinctive conditions (e.g., nutrient levels, dissolved oxygen, salinity, and temperature) at their outlets, and this study focused on their thermal function. The significance of ambient or local lagoon temperature changes may be contextualized by species-specific temperature thresholds related to metabolic activity and survival. Optimal temperature for giant Irish moss is likely between 8



to 20°C (Bird et al., 1979; Mathieson & Burns, 1971; Tasende & Fraga, 1992), and temperatures above 30°C are highly detrimental (Kübler & Davison, 1993; Lüning et al., 1986). Furthermore, blue mussels (*Mytilus edulis*) provide essential anchorage to giant Irish moss (DFO, 2009; Joseph et al., 2021), and water temperatures between 25-33°C may encumber their growth and resilience to predation (Dowd & Somero, 2013). Increasing lagoon temperatures may also be anticipated to alter primary production and macroalgae bloom dynamics (Wells et al., 2020), as well as species distributions and interactions

(Anderson, 2013). Consequently, warming of aquifers, and thus springs and groundwater-dependent streams, could negatively impact thermally vulnerable species, as mixing of groundwater into the lagoon results in lower summertime water temperatures at least locally and at low tide (Figs. 4 and S8). Also, fish have been observed aggregating in these cold-water plumes during warm days, suggesting that they are being used as refuges for thermally stressed aquatic species. Even with the groundwater warming presented in Table 2 and Fig. 9, discrete cold-water plumes will still be evident at the mouths of these springs in a

warmer climate. However, in general, for a given spring and point in time, the plume volume under key temperature thresholds will be reduced by the multi-decadal warming in the aquifer and, presumably, the lagoon.

## 6 Summary and conclusions

This study used hydrologic and thermal monitoring, groundwater tracers (temperature and radon), and numerical modelling to explore groundwater discharge and its role in maintaining survivable temperatures for the threatened ecosystem in the Basin

Head Marine Protected Area. The cold-water plume areas as revealed in drone-based thermal imagery were used to extrapolate the flow from three gauged springs to 31 ungauged springs, and the cumulative spring inflow (0.047 m$^3$ s$^{-1}$) estimated from this empirical approach was comparable to the total groundwater inflow (focused and diffuse, 0.09 m$^3$ s$^{-1}$) yielded from a $^{222}$Rn mass balance. The results also revealed that the total spring flow was comparable to the total streamflow, suggesting that, at least at a local level, springs can provide an important pathway for delivering water and energy to coastal zones.


A subsurface heat transfer model was employed to investigate the groundwater thermal sensitivity to seasonal cycles and multi-decadal climate change. The seasonal temperature amplitudes simulated at different depths for the historical period were compared to measured seasonal amplitudes from *in-situ* spring monitoring, and this comparison indicated that the lagoon intertidal springs are sourced from a range of aquifer depths (from 4 m to more than 12 m). The response to seasonal forcing

provided qualitative insight into how different springs may respond to multi-decadal forcing. Downscaled climate scenarios were used to drive future simulations to 2100, and the results revealed depth-dependent groundwater warming, with warming more pronounced at shallower depths (e.g., ≤ 2.23°C at 4.2 m) and less pronounced at greater depths (≤ 1.62°C warming at 13.9 m). The reduced warming with depth is a result of the depth-dependent lag between surface and groundwater warming signals. To our knowledge, no previous studies have investigated groundwater thermal sensitivity as a driver of future change

in coastal lagoon ecosystems. Our results indicate that submarine or intertidal groundwater discharge sourced from shallow aquifers will likely experience non-negligible warming in this century. The interaction of spring discharge warming with





lagoon changes due to sea-level rise and changing atmospheric forcing warrant further consideration and should be considered in future research using coupled thermal and hydrodynamic modelling for the lagoon.

**Author contribution**

JKA and BK designed the field program, and JKA led the execution of the field program and associated data analysis. AM assisted with radon data collection, and JT led the radon data analysis. JKA led the numerical modelling work with technical support from AM, BK, and SD. BK and RJ led the funding acquisition. BK supervised all aspects of the study. All authors contributed to the study methodology development and manuscript writing.


**Competing interests**

The authors declare that they have no conflict of interest

**Data availability**

Field data presented in this study and SHAW model input and executable files are temporarily available via a Scholars Portal Dataverse database: https://dataverse.scholarsportal.info/privateurl.xhtml?token=e27331b7-02e5-445b-a190-8cc82c4a4cd2. If accepted, this dataset will be permanently archived with a DOI, and this temporary link will be broken (refer to final paper). A readme file explains each file and how they are connected.

Other supporting tables and figures are provided in the electronic supplement to this paper.

**Acknowledgements**

Research funding was provided by an NSERC Discovery Grant to B. Kurylyk and the Ocean Frontier Institute (Opportunity Fund program) through an award from the Canada First Research Excellence Fund (CFREF). This study is also a component of an affiliate project with the Global Water Futures CFREF program. Fisheries and Oceans Canada and Souris Fish and
Wildlife are thanked for logistical, financial, and/or field support. J. KarisAllen was supported through an NSERC Canada Graduate Scholarship and the NSERC CREATE ASPIRE program. B. Kurylyk and R. Jamieson are supported through the Canada Research Chairs Program. J. Tamborski and A. Mohammed were funded through the Ocean Frontier Institute International Postdoctoral Fellowship Program while at Dalhousie University.

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



**Figure 1: (a) Location of Basin Head lagoon within Atlantic Canada. (b) Instrument, radon sampling, and identified spring locations within Basin Head lagoon and watershed over the duration of the study. Temperature sensors installed in the northeast arm of the lagoon channel were in pairs (labelled as '×2'): one at the top (affixed to a buoy) and bottom (affixed to an anchor) of the water column. (c) Enlarged view of the densely instrumented area designated by the blue box in (b). CTD = conductivity, temperature, depth. Basemap is attributed to Esri, HERE, Garmin, FAO, NOAA, USGS, © OpenStreetMap contributors, and the GIS User Community.**



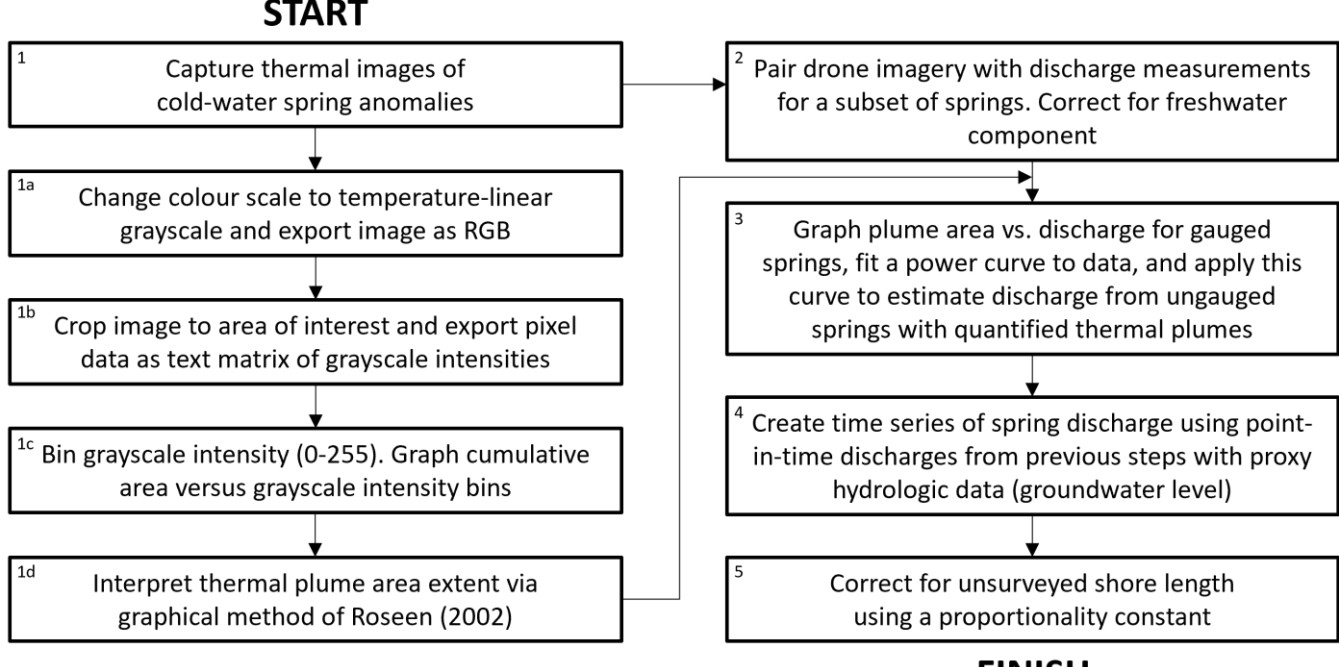

**Figure 2: Summary workflow of the spring discharge assessment technique applied in Basin Head lagoon using thermal imagery. Panels a, b, c, and d of Fig. 5 correspond with box numbers 1, 1b, 1c, and 3, respectively.**



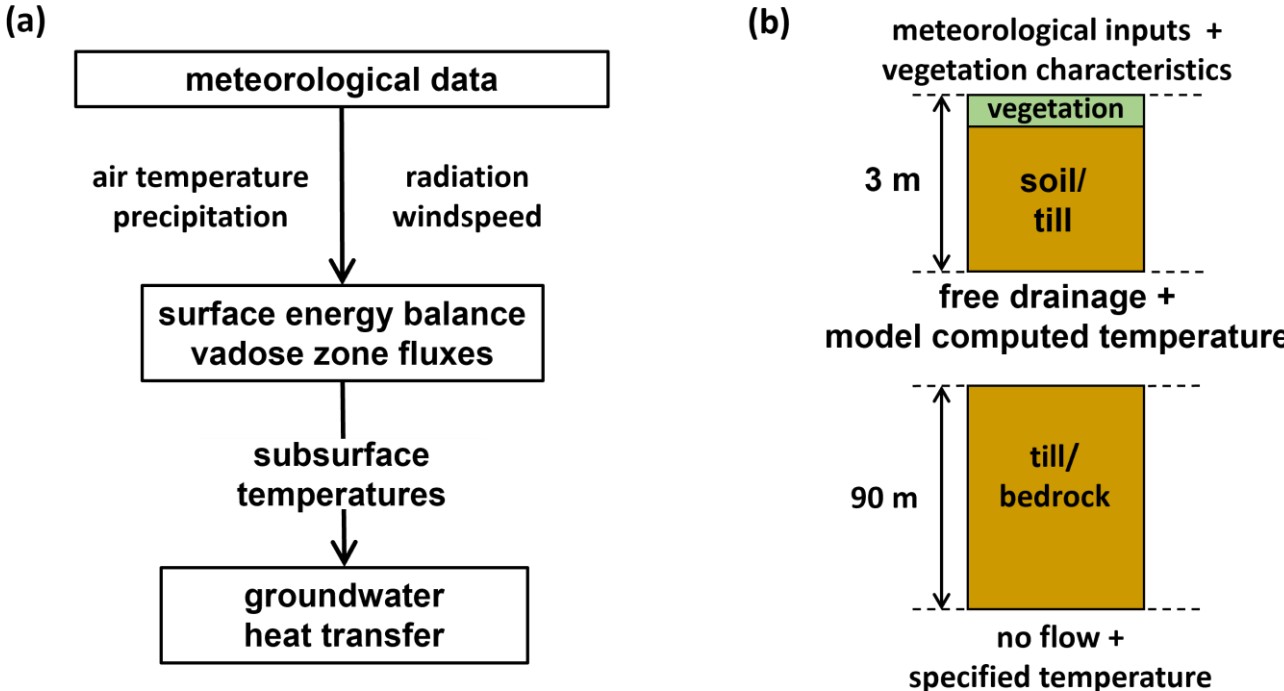

Figure 3: (a) Flowchart showing the conceptualisation of the modeling approach used in this study and (b) conceptual diagram of SHAW model set-up and boundary conditions (not drawn to scale).






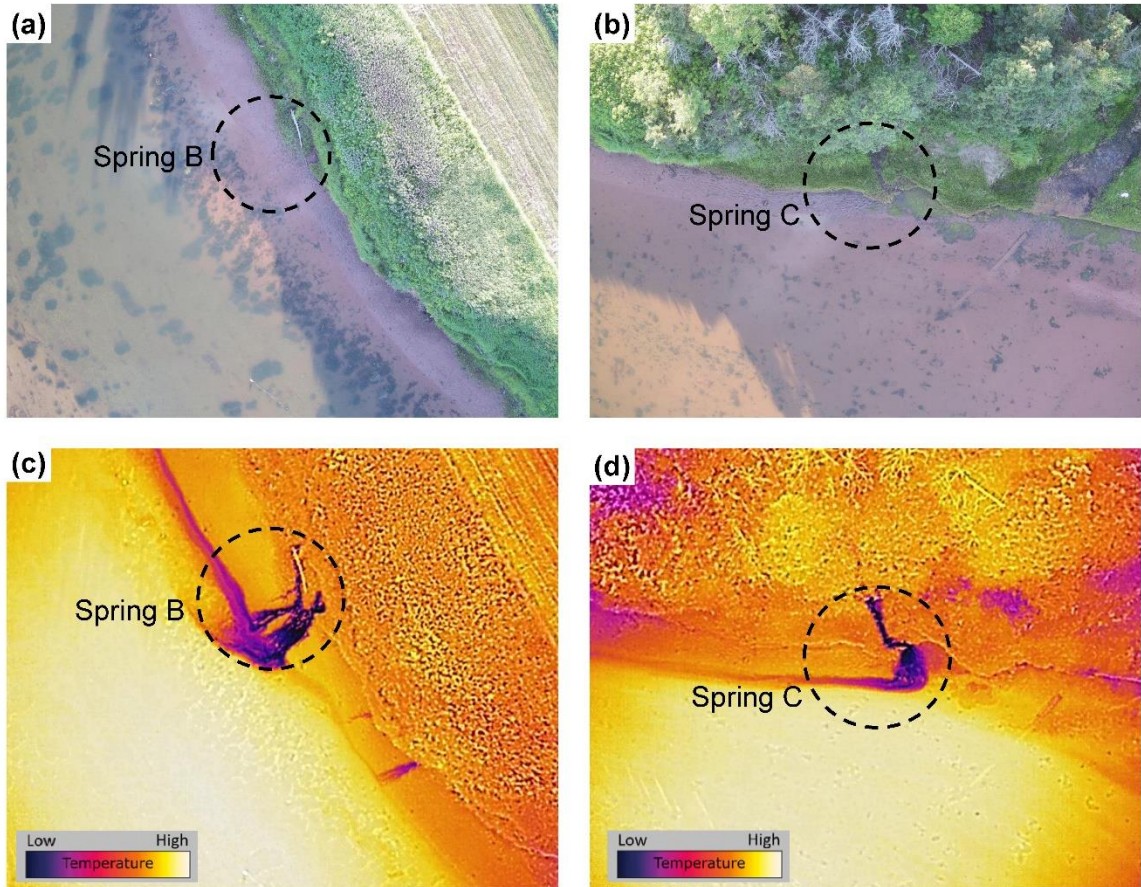

**Figure 4: Top row [(a) and (b)], visual drone images of two of the springs that were manually gauged (Springs B and C, see Table S2). Bottom row [(c) and (c)]: corresponding thermal images from the drone's thermal sensor. Scales are not equal among panels: there was a maximum thermal offset of 16°C and 12 °C between the spring water and receiving environment for (c) and (d), respectively.**




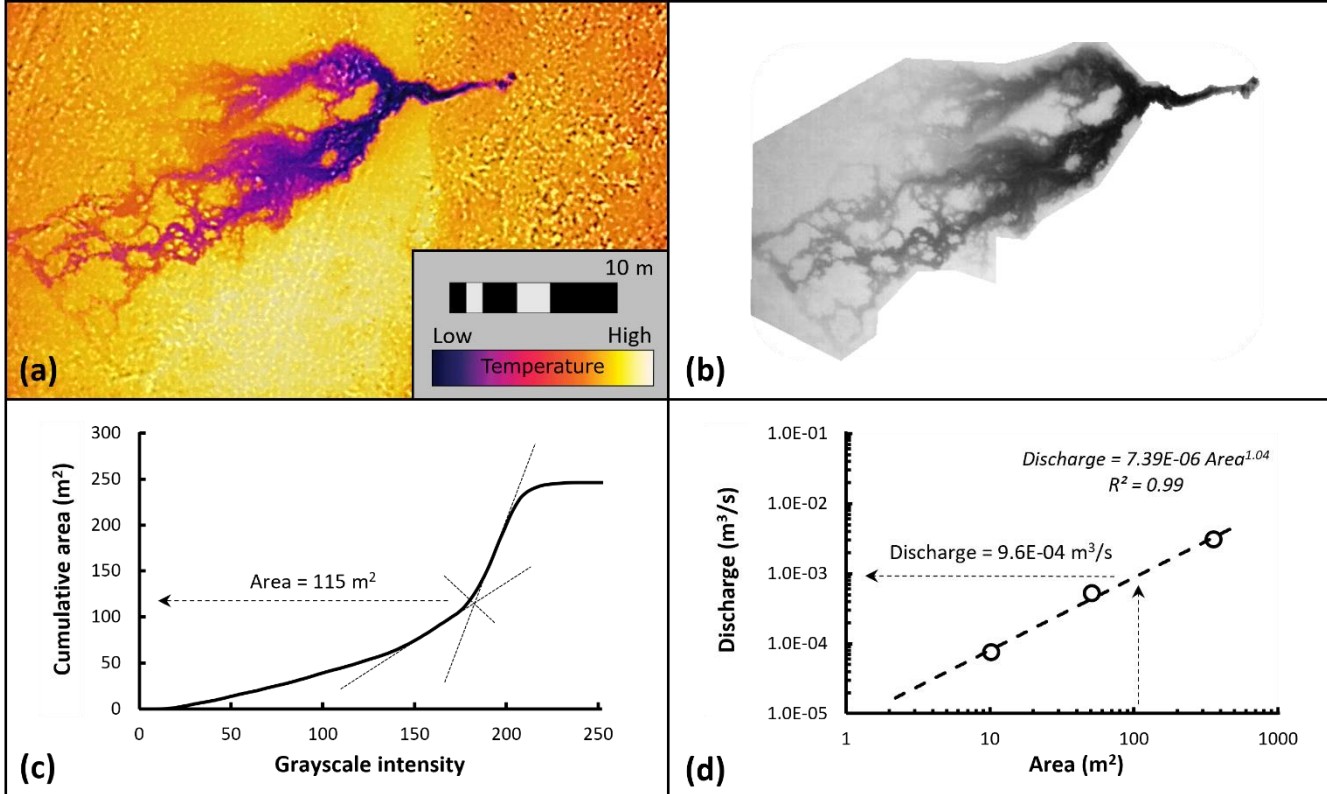

**Figure 5: Simplified workflow and results describing the area and discharge analyses of Spring 8 (included in Table S2 and Fig. S4) using the Basin Head plume size-spring discharge relationship. (a) Raw thermal image of Spring 8 cropped (rectangular) to the spring area (maximum offset of 14°C between the spring water and discharge environment). (b) Thermal image converted to 8-bit grayscale and cropped (polygonal) to thermal groups of interest. (c) Graph of thermal image pixel data in terms of cumulative area and binned grayscale values. The graphical analysis method of Roseen (2002) guided by manual inspection of image pixel values, was used to define the plume area (~115 m²). (d) The plume size-spring discharge relationship from the three gauged springs of the lagoon is used to define spring instantaneous discharge based on plume area defined in (c). Panels a, b, c, and d in this figure correspond with box numbers 1, 1b, 1c, and 3, respectively for Fig. 2.**



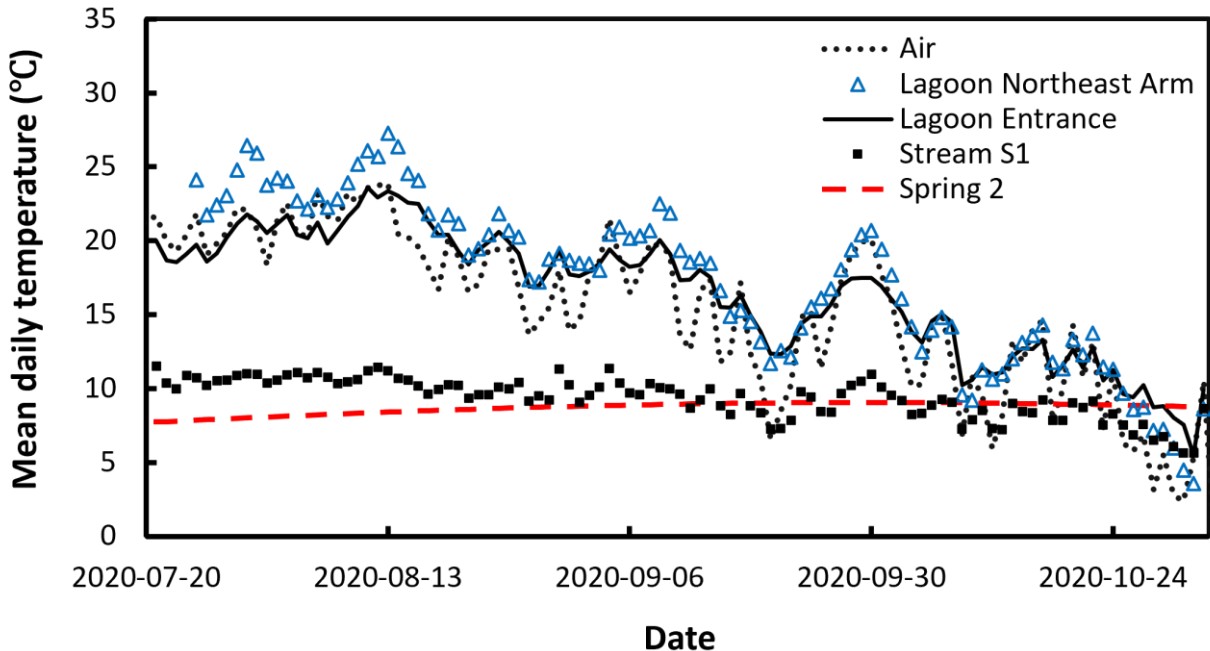

**Figure 6: Illustrative examples (subset of monitored locations) of mean daily water temperatures vs. date (yyyyy-mm-dd) for two locations in the Basin Head lagoon (i.e., entrance and northeast arm), Stream S1, and Spring 2 (with tidal effects corrected by considering the temperature only at low tide, see Fig. 7) as well as mean daily air temperature over the final four months of the study period. The lagoon northeast arm water temperature series was calculated from the average of two paired sensors (one at the lagoon water surface and the other at the channel bottom, see Fig. 1). The raw, uncorrected data and inferred annual groundwater temperature signal for Spring 2 is featured in Fig. 7c. Hourly data are in Fig. S8.**




**Figure 7: Temperature data (black) from the mouths of (a) Spring 5, (b) Spring 21, and (c) Spring 2 (see Table S2 for locations) vs. date (yyyy-mm-dd) from the Basin Head lagoon 2020 field investigations. The fitted annual temperature sine wave (GWT; in red) has a distinguishable amplitude in Springs 21 and 2 but not in Spring 5. GWT = annual groundwater temperature waveform and t = time in days.**





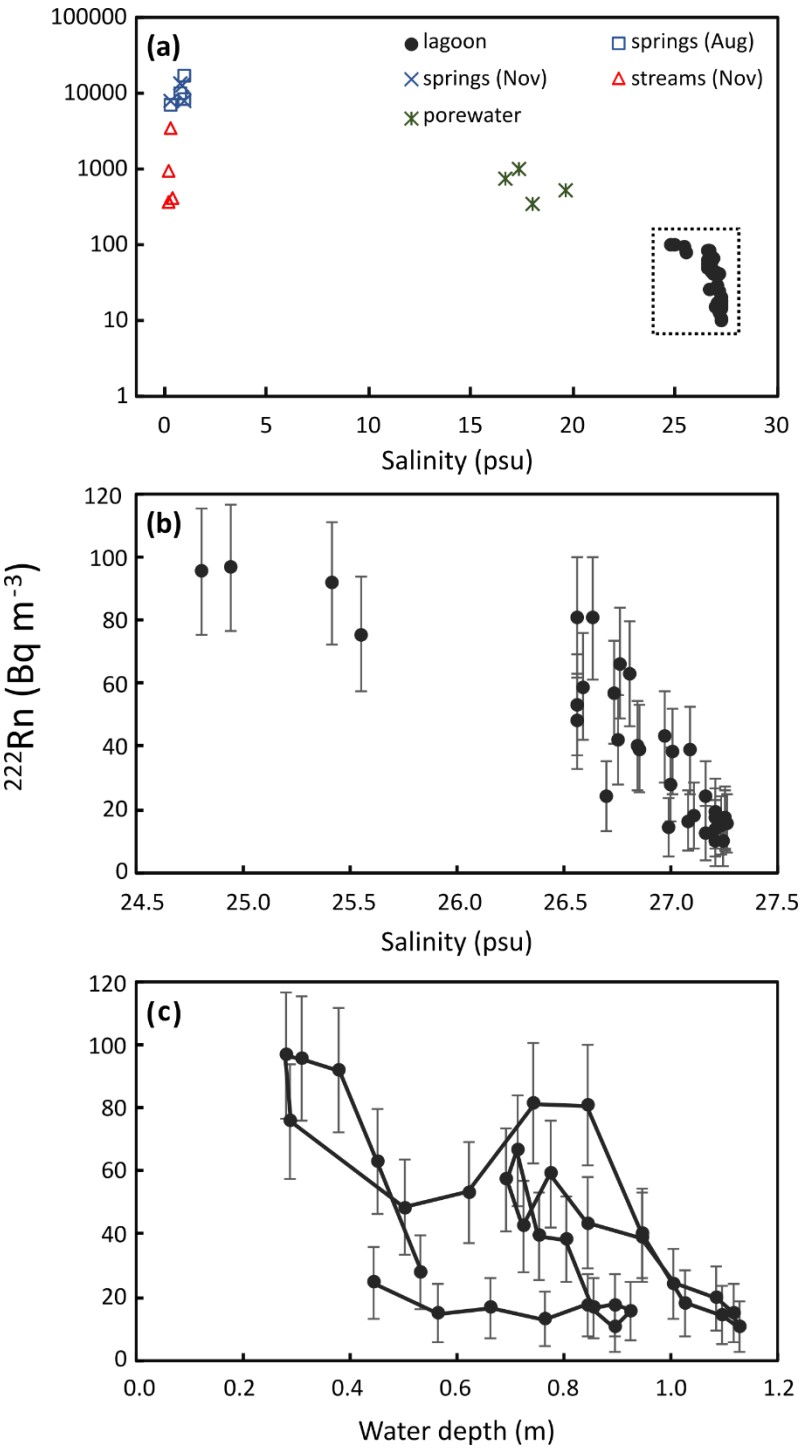


**Figure 8:** $^{222}$Rn variability versus salinity (a and b) and tidal water level (c), including hysteresis loops over two August 2020 tidal cycles; panel (b) depicts the lagoon data points outlined in (a) at a greater resolution. $^{222}$Rn values are listed in Table S3.





**Figure 9: Modelled 365-day-averaged subsurface temperatures (lines) and their associated intra-annual range (area) at two depths: (a) 4.2 m and (b) 13.9 m (representing the groundwater temperature sensor depths in our piezometer and the provincial monitoring well, respectively). The historical period (1984-2020) uses the CNRM-CR5 simulation data, and four future simulations were run for the period of 2020-2100. The beginning of the historical simulation involves a period of model domain stabilization.**





Figure 10: Historical simulation data for the years of 2016-2020 extracted from SHAW. (a) Maximum and minimum daily air
temperature and total rainfall input to the model. (b) Subsurface temperatures at various depths in response to surface forcing. The
temperature data at depths of 1 and 3 m were extracted from the surface domain, whereas the others are from the lower domain.
These modelled amplitudes may be compared to measured spring signals to estimate their source depths.







**Table 1: Basin Head lagoon heat fluxes associated with three advective processes and downwelling shortwave radiation applied across the lagoon surface area. All heat budget components are over the 35-day focused study period. Positive values indicate an addition of sensible energy to the lagoon, while negative values indicate a cooling effect. Lagoon water temperature was approximated as its median value (22°C) to calculate the advective terms (Eq. 1).**

| Heat budget component | 35-day net heat contribution | 35-day net water volume (m³) | Approx. mean water temperature (°C) |
|---|---|---|---|
| Springs | $-7.60 \times 10^{12}$ J ($-2.51 \times 10^6$ W) | 142,000 | 8 |
| Streams | $-7.67 \times 10^{12}$ J ($-2.53 \times 10^6$ W) | 153,000 | 10 |
| Rainfall | $-2.76 \times 10^{11}$ J ($-8.83 \times 10^4$ W) | 22,000 | 19 |
| Downward shortwave radiation | $3.89 \times 10^{14}$ J ($1.29 \times 10^8$ W) | NA | NA |


**Table 2: Simulated groundwater temperatures for the future SHAW simulations at the two studied depths (4.2 m = piezometer sensor depth, while 13.9 m = depth from provincial monitoring well sensor, see text). GCM = Global Circulation Model; RCP = Representative Concentration Pathway.**

| GCM | RCP | Depth (m) | Average annual temperatures (°C) | Projected change (°C)[a] |
|---|---|---|---|---|
| CNRM-CR5 | Historic | 4.2 | 7.45 – 7.80 | NA |
| CNRM-CR5 | 4.5 | 4.2 | 7.88 – 8.45 | 0.08 – 1.00 |
| CNRM-CR5 | 8.5 | 4.2 | 8.59 – 9.62 | 0.79 – 2.17 |
| MRI-CGCM3 | 4.5 | 4.2 | 7.90 – 8.61 | 0.10 – 1.16 |
| MRI-CGCM3 | 8.5 | 4.2 | 9.13 – 9.68 | 1.33 – 2.23 |
| CNRM-CR5 | Historic | 13.9 | 7.61 – 7.63 | NA |
| CNRM-CR5 | 4.5 | 13.9 | 8.26 – 8.41 | 0.63 – 0.80 |
| CNRM-CR5 | 8.5 | 13.9 | 8.79 – 9.03 | 1.16 – 1.42 |
| MRI-CGCM3 | 4.5 | 13.9 | 8.08 – 8.25 | 0.45 – 0.64 |
| MRI-CGCM3 | 8.5 | 13.9 | 9.14 – 9.23 | 1.51 – 1.62 |

[a] *The projected temperature change was calculated by comparing the last five years of the*

*future simulation to the last five years of the historic simulation.*