# Peer review of "Present and future thermal regimes of intertidal groundwater springs in a threatened coastal ecosystem"

_Hydrology and Earth System Sciences, 2022_

## Referee Comment (RC2)

Reviewer manuscript #hess-2022-49 titled "Intertidal spring discharge to a coastal ecosystem and impacts of climate change on future groundwater temperature: A multi-method investigation" by KarisAllen et al.

**General Comments:**

This manuscript describes the thermal effects of intertidal springs on coastal waters and the thermal sensitivity of these springs to climate change. Methods that used including hydrologic and thermal monitoring, groundwater tracers (temperature and radon), and numerical simulation methods. It includes an intensive work. The paper is logically clear, and the results are well discussed and explained. I have the following comments that needed to be further addressed.

1. The application of thermal information to indicate groundwater discharge has been investigated for several years. A combination of Radon, thermal images and models are not creative, therefore, it is very important to state out what are the new findings of this work?

The same, as you are combing several methods, it is better to present a more clear graph abstract or figure to show the function of each method in your study. What are their contributions in this work. The figure 2 in the current version is not that straightforward and kind of confusing.

2. Based on your data, the influence on coastal waters in the study area should be discussed in details as this is your main research goal.

3. Thermal sensitivity analysis is your another proposed research goal. As to sensitivity, you have to first clarify what this term represent in your study case? what is the difference from your model "sensitivity"? What do you mean by using this term? A factor analysis by indicating which factor is the most important to impact the thermal variation? Or is it a case to study the response of thermal change to the climate change? I am a little confused from your analysis. By the way, the data you proposed is within a short period, how this validate a long term prediction in many years?

4. A model calibration figure should be better added to show the model accuracy with continuous time series data for the main variables.

5. Some of the cited papers are not well formatted, please check them carefully.

6. In line 141, why the spring discharge is assumed to vary linearly with the piezometer water table? Whether there is any basis to confirm the rationality of the hypothesis. If yes, please add the corresponding description.

7. How to use thermal image to determine spring discharge is always a challenge as the pictures are two dimensional and your discharge is a three dimensional volume. Meanwhile, they are varied with time in every minute, and make it hard to say what you photoed can indicate more information in different hydrological period, like in the wet or dry season.

1) Please add your flying area of the drone into your location map. It can help you to show whether they are consistent with the Radon data and you know the drone has a limitation to cover large area within a short time period.

2) In lines 275-280, three springs were selected to determine the power function relationship between spring discharge and thermal plume area for the lagoon. There are about 40 springs in this area. Are the three springs representative? In addition, are the three data points too little to yield the mathematical relationship between the two?

3) In line 281, the area of the spring is evaluated based on the irregular clipping of the spring location on the thermal image. What is the standard of graphic clipping? What principles need to be followed?

8. In line 253, the 1-D subsurface heat and water transport model established in the study area includes a saturated area of 3-93m. Do you have a temperature distribution along the perpendicular cross section to show the area that is effected by the spring plume. This is important to support that why the authors only select the temperature data at the depths of 1m, 3m, 5m, 10.28m, and 15.24m in the numerical modeling approach in response to the surface forcing (Fig. 10)?

9. In line 823, please change "Bottom row [(c) and (c)]" to " Bottom row [(c) and (d)]".

10. In Fig. 4(a)-4(b), please add the corresponding scale bar or pixel size of the image.

11. In Fig 10(a), the precipitation data over the years is unclear and lacks units. Please modify it.

12. The work is comprehensive, it would be a good work if the main research goal and methods, especially their connections, can be stated very clear through the paper.

---

## Author Comment (AC1)

**General author comment:** We appreciate the time spent by the AE securing reviews and the consideration of our manuscript for potential publication in HESS. We also appreciate the helpful comments from the peer reviewers. We respond to these comments on a point-by-point basis below (our replies in blue italics). We feel these changes will improve our paper. At a high level, the revised manuscript will do a better job connecting our field data interpretation and modeling (present and future conditions). As a general response to some comments below, we note that (1) groundwater thermal modeling is far more robust than groundwater flow modeling or shallow soil temperature modeling (flashy signals are modulated) and (2) our focus in the modeling is to investigate long-term system response (sensitivity) to seasonal and decadal forcing rather than to exactly reproduce the conditions at our field site. This will be better articulated in our revised manuscript.

**Reviewer 1**

**Overview:** Thermal impacts of springs on coastal waters and the sensitivity of these springs to climate change are not well understood. To address this issue, this study used field study for a threatened coastal lagoon ecosystem in south-eastern Canada by pairing in-situ thermal and drone-based thermal imagery monitoring to estimate the discharge to the lagoon. It also applied a numerical model to relate measured spring temperatures to their respective aquifer depths, and to study long-term groundwater warming. The value of this study lies on providing some insights to coastal ecosystem management. I have some comments that may improve the quality of this article. Please see the detail as follows:

*Thank for this accurate overview of our manuscript and your time in helping us improve our work.*

**Comment 1:** There are two parts of this study: analyzing measurements and numerical modeling. I think the link between the two parts is that the model was employed to match the measurements to locate the aquifer depth that provides the water source to the lagoon. However, this link is not stressed in the text, so it looks like two separate studies. Most importantly, the major aim of the modeling (i.e., studying the sensitivity of groundwater temperature to climate change) is not related to the measurements analysis. I think the authors should work on the text more to link these parts to make them integrated.

*This is a good point. The goal of the study was to look at both the present and future thermal impacts of these springs, and thus the field work (present) and modeling (present and future) are directly related. Also, the measured spring temperatures were used to infer the spring depths, which was a key factor in our numerical model (the linkage the reviewer alludes to above). However, we agree that these concepts should be tied together more closely in the text, and we will modify the introduction and methods to highlight this. We will also add a new methods figure that shows how the different aspects of the study (hydrology, drone sensing, radon, and numerical modeling) are integrated. We believe this will result in much stronger messaging and overall scientific narrative.*

**Comment 2**: In regards to the hydrological modeling, some necessary uncertainty analyses is missing. Although two data sets of forcings were used, the assumptions and deficits of the hydrological model SHAW were not introduced and the related uncertainties or bias that may be derived from them were not analyzed. The authors need to discuss the uncertainties from many aspects (e.g., model, data, assumptions) and their possible influences to the results in the text to add the value of this manuscript.

*In general, thermal modeling of hydrogeologic systems is far more robust than hydrogeologic modeling (e.g. water flux or head modeling) simply because the associated parameters (thermal properties) are far more constrained. This is particularly true below the shallow soil zone experiencing diel temperature fluctuations. However, we agree that adding another paragraph to the discussion text for the modeling will help with acknowledging some of the assumptions and uncertainty in the modeling approach. We will also refer to the rich literature on SHAW applications/limitations as this is one of the most commonly applied ground temperature models.*

**Comment 3:** L105, "methods section": Too many words were used to introduce the monitoring software and system in section 3.1 and 3.2 which I think is not very relevant to the scientific topic. Is it possible simplify those sections and move some of the contents to SI?

*Agreed – we will condense this text and move any tertiary points to the supplement. This is not fundamentally a study on thermal image analysis; rather that was just a step in our methods.*

**Comment 4:** L230-231, "The conceptual … heat transport processes.": Please introduce more about the water and heat transport model. What key transport processes the model preserved?

*We will add a couple more sentence on the surface energy balance and subsurface heat fluxes (conduction and advection) in the model. We will also add the governing subsurface heat transfer PDE in the main text or in the supplement. We will also emphasize this is a standard model.*

**Comment 5:** L246-247, "A detailed description…detailed in Flerchinger (2017).": As mentioned above, a bit more about the SHAW model could be introduced in the text, rather than just refering another paper.

*See above.*

**Comment 6:** L258, "a daily resolution": Most land models use 1800s as the timestep. Is it a daily resolution too coarse for the soil moisture simulation?

*This time step is pretty typical in groundwater temperature modeling (e.g. Langford et al., 2020, Groundwater) when sub-daily soil moisture and temperature fluctuations are not of interest (we are looking at more modulated seasonal or decadal signals). Although soil moisture plays a secondary role (e.g. in altering soil thermal properties) we do not need to resolve these changes at a high frequency. We will add one sentence indicating our justification for this time step.*

**Comment 7:** L260-261, "The minimum and … RCP4.5 hindcast model": Why didn't use the historical reanalysis dataset as forcings? It would be more accurate than the model outputs.

*It is a valid point that we could provide more details in the paper on our rationale for the dataset selection for the historic period forcing, although the cited Warner study reveals reasonable agreement between the two datasets. We will add text in the methods section for this point and explain our direction. It's worth noting that we do not have a direct long-term climate record in Basin Head, and that there are issues with either dataset given the reanalysis/statistical downscaling/modeling employed. Also, for our assessment we are not trying to reproduce or compare daily conditions but rather multi-year averages in modulated groundwater temperatures (see Table 2), and thus higher-frequency discrepancies between forcing data are less relevant for our modeling purposes.*

**Comment 8:** L269-270, "(1) CNRM-CM5 … MRI-CGCM3, RCP8.5": What are the spatial resolutions for these model outputs and reanalysis data?

*We will add more details to the revised manuscript. This location was taken within a ~10 x 6 km grid. We used the BCCAQv2 statistically downscaled data which is roughly 10x10km (downscaled from 100km).*

*https://climatedata.ca/explore/location/?loc=BAAHA&location-select-temperature=tx_mean&location-select-precipitation=prcptot&location-select-other=ice_days*

**Comment 9:** L279-280, "The paired discharge … relationship for the lagoon.": I don't think this linear relationship is reliable enough based on only three sites.

*Thank you for this comment. We agree that having three points is not ideal, and have noted this in the text (see L410). However, we do have a couple thoughts in reply to this:*

1) *The relationship is not linear – it's a power relationship that appears linear on a log plot (see L280)*
2) *The relationship between plume size and flow rate is only valid for a single weather condition and tidal level. Also, many springs are only exposed for a short period at low tide. Thus, all of these points must be taken concurrently, and require flumes to be set up in each spring. Even with a large field team (about 6 people), we were only able to accurately gauge 3 springs at the same low tide point. Also, this event presented very ideal conditions (heat wave maximizing the thermal contrast, coincident with spring low tide exposing the most springs). If we were to get more points, we would have to return with a larger team for an entire new field campaign with very little chance of having the same ideal conditions. Thus, we do not think that collecting more data points to improve this relationship is feasible. Also, we are mostly interpolating (rather than extrapolating) with our plume Area-Q relationship as all but one spring had a smaller plume area than the largest spring we gauged (see supplement table). Thus, we think our approach is reasonable as a first-order assessment. We will add a couple more sentences highlighting the challenges of collecting more data points for this relationship, the unique area-Q relationship for given tide and weather conditions, the ideal environmental conditions during our study, and the limitations of our approach.*

---

## Author Comment (AC2)

**Reviewer 2**

General Comments : This manuscript describes the thermal effects of intertidal springs on coastal waters and the thermal sensitivity of these springs to climate change. Methods that used including hydrologic and thermal monitoring, groundwater tracers (temperature and radon), and numerical simulation methods. It includes an intensive work. The paper is logically clear, and the results are well discussed and explained. I have the following comments that needed to be further addressed.

*Thank you for considering our manuscript and providing this overview of our study, which is a good summary of our work.*

**Comment 1**: The application of thermal information to indicate groundwater discharge has been investigated for several years. A combination of Radon, thermal images and models are not creative, therefore, it is very important to state out what are the new findings of this work? The same, as you are combing several methods, it is better to present a more clear graph abstract or figure to show the function of each method in your study. What are their contributions in this work. The figure 2 in the current version is not that straightforward and kind of confusing.

*We aren't sure if the reviewer's comment about combining radon, thermal images and models lacking creativity is (1) suggesting there are other studies that combine these methods (we reviewed the literature extensively and are not aware of any that integrate thermal sensing, radon analysis and numerical modeling) or (2) stating that merely combining these methods is not enough for a novel paper. If the latter, we agree. The novel and important contributions of this study relate to better understanding the present and future thermal (and to a lesser degree hydrologic) function of intertidal springs for warming coastal ecosystems (see L80). Nevertheless, we agree that the scientific narrative could be improved somewhat, and we will update the introduction, methods, and conclusions to better elucidate our key objectives, describe how our methods tie together, and emphasize the key outcomes of the study. We will also slightly modify the title to focus more on 'present and future thermal influence of intertidal springs in coastal ecosystems'.*

*The comment in reference to Figure 2 is likely pointing out that the **overall** methodology is not clear in Figure 2 (which only focuses on the thermal image analysis). We will add a new figure that integrates all of our methodological approaches and shows the relationships between them.*

**Comment 2**: Based on your data, the influence on coastal waters in the study area should be discussed in details as this is your main research goal.

*We are slightly confused by this comment as the influence of the springs on the coastal waters is described in sections 5.1, 5.2, and 5.4; indeed it is the focus of these discussion sections. Perhaps the reviewer is looking for more focus on coastal zone processes that would differentiate this from inland studies. We do mention tidal impacts on the thermal plume dynamics (section 5.1), tidal impacts on the energy exchange (5.2). Also, the ecosystem we focus on in section 5.4 is distinctly coastal. We will add additional text in section 5.4 on the influence of water*

*temperature on blue mussels (which clump with the Giant Irish Moss) at the study site and the implications for management in this Marine Protected Area.*

**Comment 3**: Thermal sensitivity analysis is your another proposed research goal. As to sensitivity, you have to first clarify what this term represent in your study case? what is the difference from your model "sensitivity"? What do you mean by using this term? A factor analysis by indicating which factor is the most important to impact the thermal variation? Or is it a case to study the response of thermal change to the climate change? I am a little confused from your analysis. By the way, the data you proposed is within a short period, how this validate a long term prediction in many years?

*Thermal sensitivity is a term used in aquatic thermal regime work to refer to the change in water temperature divided by atmospheric forcing (often a change in air temperature). We will define this term more specifically in the modified text. This is completely distinct from model sensitivity.*

*With regard to the short term vs. long term issue: numerical models in hydrology and hydrogeology often use data from a short period to calibrate or assess a model and then apply that model to forward model decades into the future under some downscaled climate scenario or scenarios. This is not unique to our study. Being able to reproduce seasonal groundwater temperature signals under seasonal forcing does indicate that our general thermal properties and model thermal response is reasonably established, particularly given the strong physical basis for SHAW. We will, however, add one sentence to our discuss on the differences between modeling seasonal and decadal groundwater temperatures (a key point in Kurylyk et al. 2015 HESS) and thus acknowledge the limitation of this work.*

**Comment 4:** A model calibration figure should be better added to show the model accuracy with continuous time series data for the main variables.

*We will add new text or a sub-table discussing a comparison between the measured/modeled groundwater temperature means and amplitudes.*

**Comment 5:** Some of the cited papers are not well formatted, please check them carefully.

*Thank you for noting this; we will fix all issues with in-text citations and associated references.*

**Comment 6:** In line 141, why the spring discharge is assumed to vary linearly with the piezometer water table? Whether there is any basis to confirm the rationality of the hypothesis. If yes, please add the corresponding description.

*This is just based on Darcy's law (Q= kiA). The piezometer water level indicates the groundwater head, and thus is a reflection of the aquifer-lagoon hydraulic gradient averaged across the tidal cycle. We will edit this text slightly to make this clearer.*

**Comment 7:** How to use thermal image to determine spring discharge is always a challenge as the pictures are two dimensional and your discharge is a three dimensional volume. Meanwhile,

they are varied with time in every minute, and make it hard to say what you photoed can indicate more information in different hydrological period, like in the wet or dry season.

*We agree with these noted challenges; please see our specific responses below.*

**Sub-comment 1)** Please add your flying area of the drone into your location map. It can help you to show whether they are consistent with the Radon data and you know the drone has a limitation to cover large area within a short time period.

*The flying area is not intended to be consistent with radon data. The reason for this is that the springs can only be found in the fractured sandstone on one side of the lagoon, whereas the radon data integrates the groundwater influence across the lagoon. Nevertheless, we will add the approximate flying area to a map in our revised manuscript as we do think this is a good idea that will help the reader visualize the process.*

**Sub-comment 2)** In lines 275-280, three springs were selected to determine the power function relationship between spring discharge and thermal plume area for the lagoon. There are about 40 springs in this area. Are the three springs representative? In addition, are the three data points too little to yield the mathematical relationship between the two?

*This is a very reasonable concern. Please see our response to reviewer 1, comment 9 to explain why more points are not possible. We will explain this in more detail and expand discussion on limitations in the revised text.*

**Sub-comment 3)** In line 281, the area of the spring is evaluated based on the irregular clipping of the spring location on the thermal image. What is the standard of graphic clipping? What principles need to be followed?

*We will add a few sentences to the supplement to describe our clipping process. There is no standard clipping approach in the literature, but the key point is to be consistent to allow comparison across the dataset.*

*Irregular clipping was conducted, where possible, to isolate two distinct thermal groups and the transition between them (the lagoon water, and the spring water). The main priority was to reduce interference from thermal groups with overlapping temperature ranges (e.g., foliage, shoreline). Rectangular cropping enables too many sources of thermal interference, which in several cases erroneously altered the calculated area.*

**Comment 8:** In line 253, the 1-D subsurface heat and water transport model established in the study area includes a saturated area of 3-93m. Do you have a temperature distribution along the perpendicular cross section to show the area that is effected by the spring plume. This is important to support that why the authors only select the temperature data at the depths of 1m, 3m, 5m, 10.28m, and 15.24m in the numerical modelingapproach in response to the surface forcing (Fig. 10)?

*The only groundwater temperature data we have are in the coastal piezometer and upland well as described in the manuscript. We do not have data revealing the temperature distribution down*

*to 93 m. We extend the model far below our depth of interest to remove any effects of geothermal heat flux as is common in such modeling, and will explain this in the revised text. The depths indicated here refer to standard depths to show damping and lagging (1, 3, and 5 m, Fig. 10b). The other depths (10.28 and 15.24) in Fig. 10b do give the appearance of being randomly selected, and thus will be changed to reflect time series at the calibration points to better justify the choice of depths.*

**Comment 9:** In line 823, please change "Bottom row [(c) and (c)]" to " Bottom row[(c) and(d)]".

*Thank you for catching this typo, which will be fixed in the revision.*

**Comment 10:** In Fig. 4(a)-4(b), please add the corresponding scale bar or pixel size of the image.

*Good point; this will be added in the revision.*

**Comment 11:** In Fig 10(a), the precipitation data over the years is unclear and lacks units. Please modify it.

*Thank you, somehow this must have got cut off during the image upload and manuscript compilation. We will fix this in the revised version.*

**Comment 12:** The work is comprehensive, it would be a good work if the main research goal and methods, especially their connections, can be stated very clear through the paper.

*Thank you. We feel that the modifications in response to concerns from both reviewers should help clarify the goals and methods of the study.*

---

## Author Response (AR1)

Dear Dr. Ursino:

We appreciate the time spent securing reviews and the consideration of our manuscript for potential publication in HESS. We also appreciate the helpful comments from the peer reviewers. We respond to these comments on a point-by-point basis below (our replies in blue italics). These changes have substantially improved our paper and are thankful for the benefits of the peer-review process. **All line numbers refer to the tracked changes version of the manuscript.**

At a high level, the revised manuscript does a better job connecting our field data interpretation and modeling (present and future conditions). We added text in several locations for this purpose as well as a new figure (2) showing how the elements are combined. This replaces an older methods figure (formerly Figure 2) that we have moved to the supplement (Figure S1). We have switched the order of Figures 9 and 10 (and associated text) to discuss seasonal signals before climate change signals. We also emphasize the novelty and scientific goals of the paper better in our introduction and conclusion and have trimmed our methods in places to reduce the word count (given the suggested additions in other locations). As a general response to some comments below, we note that (1) groundwater thermal modeling is far more robust than groundwater flow modeling or shallow soil temperature modeling (flashy signals are modulated) and (2) our focus in the modeling is to investigate long-term system response (sensitivity) to seasonal and decadal forcing rather than to exactly reproduce the daily conditions at our field site. This is all better articulated in the revised manuscript. Despite the improved presentation, no scientific findings have been changed since the first manuscript version. We hope the present version is suitable for publication in *HESS*.

Sincerely,

Jason KarisAllen and Barret Kurylyk for all authors

**Reviewer 1**

**Overview:** Thermal impacts of springs on coastal waters and the sensitivity of these springs to climate change are not well understood. To address this issue, this study used field study for a threatened coastal lagoon ecosystem in south-eastern Canada by pairing in-situ thermal and drone-based thermal imagery monitoring to estimate the discharge to the lagoon. It also applied a numerical model to relate measured spring temperatures to their respective aquifer depths, and to study long-term groundwater warming. The value of this study lies on providing some insights to coastal ecosystem management. I have some comments that may improve the quality of this article. Please see the detail as follows:

*Thank for this accurate overview of our manuscript and your time in helping us improve our work.*

**Comment 1:** There are two parts of this study: analyzing measurements and numerical modeling. I think the link between the two parts is that the model was employed to match the measurements to locate the aquifer depth that provides the water source to the lagoon. However, this link is not stressed in the text, so it looks like two separate studies. Most importantly, the major aim of the modeling (i.e., studying the sensitivity of groundwater temperature to climate change) is not related to the measurements analysis. I think the authors should work on the text more to link these parts to make them integrated.

*This is a good point. The goal of the study is to look at both the present and future thermal impacts of these springs, and thus the field work (present) and modeling (present and future) are directly related. Also, the measured spring temperatures were used to infer the spring depths, which was a key factor in our numerical model (the linkage the reviewer alludes to above). However, we agree that these concepts were not tied together in our original text, and we have modified the introduction and methods to highlight this. For example, we have added "numerical model informed by field data" (L93) and "model calibrated with groundwater well data" (L101) to the introduction as well as the final introduction sentence that refers to these elements being used in concert ("Field data and numerical modeling results are collectively used", L103-105).*

*More importantly, we have added a brand new figure (Figure 2) showing these study components. While we concede this new figure is a bit busy, we think this is inevitable and overcome this by carefully explaining it in the text ((L123-137). We also use this figure as the backbone for our entire methods section, and occasionally refer back to the boxes in the figure throughout the revised methods section.*

**Comment 2**: In regards to the hydrological modeling, some necessary uncertainty analyses is missing. Although two data sets of forcings were used, the assumptions and deficits of the hydrological model SHAW were not introduced and the related uncertainties or bias that may be derived from them were not analyzed. The authors need to discuss the uncertainties from many aspects (e.g., model, data, assumptions) and their possible influences to the results in the text to add the value of this manuscript.

*In general, thermal modeling of hydrogeologic systems is far more robust than hydrogeologic modeling (e.g. water flux or head modeling) simply because the associated parameters (thermal properties) are far more constrained. This is particularly true below the shallow soil zone which experiences diel temperature fluctuations. However, we agree that adding some more text to the discussion text for the modeling (see L573-578) now helps with acknowledging some of the assumptions and uncertainty in the modeling approach. We have also referred to the fact that SHAW is a well-applied model (L311-312) for this purpose. Finally, we explain that groundwater temperature modeling is not as uncertain as, for example, soil moisture modeling (L328-330).*

**Comment 3:** L105, "methods section": Too many words were used to introduce the monitoring software and system in section 3.1 and 3.2 which I think is not very relevant to the scientific topic. Is it possible simplify those sections and move some of the contents to SI?

*We agree. We have reduced aspects of section 3.1 (several sentences moved to supplement) and section 3.2 (mostly instrument information that is already found in Table S1). This occurred in many sentences, so we encourage the reviewer and editor to skim the tracked changes in this section. We have also removed the original Figure 2 from the methods and replaced it with a more holistic Figure 2 that discusses all of the study aspects (not just the image processing).*

**Comment 4:** L230-231, "The conceptual … heat transport processes.": Please introduce more about the water and heat transport model. What key transport processes the model preserved?

*Thank you. We have added a few sentences on the thermal processes and the main pertinent PDE (ground heat transfer) used in SHAW and have emphasized this is a standard model (L299-312).*

**Comment 5:** L246-247, "A detailed description…detailed in Flerchinger (2017).": As mentioned above, a bit more about the SHAW model could be introduced in the text, rather than just refering another paper.

*See above.*

**Comment 6:** L258, "a daily resolution": Most land models use 1800s as the timestep. Is it a daily resolution too coarse for the soil moisture simulation?

*This time step is pretty typical in groundwater temperature modeling (e.g. Langford et al., 2020, Groundwater) when sub-daily soil moisture and temperature fluctuations are not of interest (we are looking at more modulated seasonal or decadal signals). Although soil moisture plays a secondary role (e.g. in altering soil thermal properties) we do not need to resolve these changes at a high frequency. We have added 2 sentences explaining this (L326-330)*

**Comment 7:** L260-261, "The minimum and … RCP4.5 hindcast model": Why didn't use the historical reanalysis dataset as forcings? It would be more accurate than the model outputs.

*It is a fair statement that we should have provided more details in the original manuscript on our rationale for the dataset selection for the historic period forcing, although the cited Warner study reveals reasonable agreement between the two datasets. We have added text to the methods section (L334-338) for this point and explain our direction. We do not have a direct long-term climate record in Basin Head, and that there are issues with either dataset given the reanalysis/statistical downscaling/modeling employed. Also, for our assessment we are not trying to reproduce or compare daily conditions but rather multi-year averages in modulated groundwater temperatures (see Table 2), and thus higher-frequency discrepancies between forcing data are less relevant for our modeling purposes.*

**Comment 8:** L269-270, "(1) CNRM-CM5 … MRI-CGCM3, RCP8.5": What are the spatial resolutions for these model outputs and reanalysis data?

*We used the BCCAQv2 statistically downscaled data which is roughly 10x10km (downscaled from 100km) as explained in the link below. We have added a short sentence to the text to explain this (L337).*

*https://climatedata.ca/explore/location/?loc=BAAHA&location-select-temperature=tx_mean&location-select-precipitation=prcptot&location-select-other=ice_days*

**Comment 9:** L279-280, "The paired discharge … relationship for the lagoon.": I don't think this linear relationship is reliable enough based on only three sites.

*Thank you for this comment. We do have a couple thoughts in reply to this:*

1) *The relationship is not linear – it's a power relationship that appears linear on a log plot*
2) *The relationship between plume size and flow rate is only valid for a single weather condition and tidal level. Also, many springs are only exposed for a short period at low tide. Thus, all of these points must be taken concurrently, and require flumes to be set up in each spring. Even with a large field team (about 6 people), we were only able to accurately gauge 3 springs at the same low tide point. Also, this event presented very ideal conditions (heat wave maximizing the thermal contrast, coincident with spring low tide exposing the most springs). If we were to get more points, we would have to return with a larger team for an entire new field campaign with very little chance of having the same ideal conditions. Thus, we do not think that collecting more data points to improve this relationship is feasible. Also, we are mostly interpolating (rather than extrapolating) with our plume Area-Q relationship as all but one spring had a smaller plume area than the largest spring we gauged (see supplement table). Thus, we think our approach is reasonable as a first-order assessment.*

*Accordingly, we have added a few sentences explaining the lack of points in our Q-area relationship but why we think the measurement timing is ideal (L203-218).*

**Reviewer 2**

General Comments : This manuscript describes the thermal effects of intertidal springs on coastal waters and the thermal sensitivity of these springs to climate change. Methods that used including hydrologic and thermal monitoring, groundwater tracers (temperature and radon), and numerical simulation methods. It includes an intensive work. The paper is logically clear, and the results are well discussed and explained. I have the following comments that needed to be further addressed.

*Thank you for considering our manuscript and providing this overview of our study, which is a good summary of our work.*

**Comment 1**: The application of thermal information to indicate groundwater discharge has been investigated for several years. A combination of Radon, thermal images and models are not creative, therefore, it is very important to state out what are the new findings of this work? The same, as you are combing several methods, it is better to present a more clear graph abstract or

figure to show the function of each method in your study. What are their contributions in this work. The figure 2 in the current version is not that straightforward and kind of confusing.

*We aren't sure if the reviewer's comment about combining radon, thermal images and models lacking creativity is (1) suggesting there are other studies that combine these methods (we reviewed the literature extensively and are not aware of any that integrate thermal sensing, radon analysis and numerical modeling) or (2) stating that merely combining these methods is not enough for a novel paper. If the latter, we agree. The novel and important contributions of this study relate to better understanding the present and future thermal (and to a lesser degree hydrologic) function of intertidal springs for warming coastal ecosystems). Nevertheless, we agree that the scientific narrative could be improved somewhat, and we have updated the intro (L90-91, 93-95, 103-105), methods (e.g., Figure 2 and surrounding text) and conclusion (see minor changes throughout conclusions), to better elucidate our key objectives, describe how our methods tie together, and emphasize the key outcomes of the study. We also modified our title to focus more on 'present and future thermal influence of intertidal springs in coastal ecosystems'.*

*The comment in reference to Figure 2 is likely pointing out that the **overall** methodology is not clear in Figure 2 (which only focuses on the thermal image analysis). We have moved this old figure 2 to the supplement and replaced it with a new Figure 2 that integrates all of our methodological approaches and shows the relationships between them.*

**Comment 2**: Based on your data, the influence on coastal waters in the study area should be discussed in details as this is your main research goal.

*We are slightly confused by this comment as the influence of the springs on the coastal waters is described in sections 5.1, 5.2, and 5.4; indeed it is the focus of these discussion sections. Perhaps the reviewer is looking for more focus on coastal zone processes that would differentiate this from inland studies. We do mention tidal impacts on the thermal plume dynamics (section 5.1), tidal impacts on the energy exchange (5.2). Also, the ecosystem we focus on in section 5.4 is distinctly coastal. While we think the 'coastal discussion' is sufficient, we have added additional text in section 5.4 on the broader impacts to coastal waters at the study site (i.e. beyond the spring mouths) (L600-603).*

**Comment 3**: Thermal sensitivity analysis is your another proposed research goal. As to sensitivity, you have to first clarify what this term represent in your study case? what is the difference from your model "sensitivity"? What do you mean by using this term? A factor analysis by indicating which factor is the most important to impact the thermal variation? Or is it a case to study the response of thermal change to the climate change? I am a little confused from your analysis. By the way, the data you proposed is within a short period, how this validate a long term prediction in many years?

*Thermal sensitivity is a term used in aquatic thermal regime work to refer to the change in water temperature divided by atmospheric forcing (often a change in air temperature). This is*

*completely distinct from model sensitivity. We have added a definition to this in our third introduction paragraph (L75-78).*

*With regard to the short term vs. long term issue: numerical models in hydrology and hydrogeology often use data from a short period to calibrate or assess a model and then apply that model to forward model decades into the future under some downscaled climate scenario or scenarios. This is not unique to our study. Being able to reproduce seasonal groundwater temperature signals under seasonal forcing does indicate that our general thermal properties and model thermal response is reasonably established, particularly given the strong physical basis for SHAW. Kurylyk et al. (2015, HESS) discuss the limitations of using short-term (e.g. seasonal) relationships to infer long-term thermal sensitivity, but those limitations only apply to statistical approaches, not process-based modeling. Hence, we don't really see this as a limitation worth noting, but we have added a couple of sentences on other limitations of our model that we consider to be more substantial (L573-578).*

**Comment 4:** A model calibration figure should be better added to show the model accuracy with continuous time series data for the main variables.

*We have added a new supplementary table (Table S5) discussing a comparison between the measured/modeled groundwater temperature means and amplitudes.*

**Comment 5:** Some of the cited papers are not well formatted, please check them carefully.

*Thank you for noting this; we have reviewed our work and have not identified very many issues with our reference list but will work with the typesetter (if accepted) to fix any minor remaining issues.*

**Comment 6:** In line 141, why the spring discharge is assumed to vary linearly with the piezometer water table? Whether there is any basis to confirm the rationality of the hypothesis. If yes, please add the corresponding description.

*This is just based on Darcy's law (Q= kiA). The piezometer water level indicates the groundwater head, and thus is a reflection of the aquifer-lagoon hydraulic gradient across the tidal cycle. We have added this the text (L177).*

**Comment 7:** How to use thermal image to determine spring discharge is always a challenge as the pictures are two dimensional and your discharge is a three dimensional volume. Meanwhile, they are varied with time in every minute, and make it hard to say what you photoed can indicate more information in different hydrological period, like in the wet or dry season.

*We agree with these noted challenges; please see our specific responses below as well as our response to reviewer 1 (comment 9) and associated textual changes.*

**Sub-comment 1)** Please add your flying area of the drone into your location map. It can help you to show whether they are consistent with the Radon data and you know the drone has a limitation to cover large area within a short time period.

*The flying area is not intended to be consistent with radon data. The reason for this is that the springs can only be found in the fractured sandstone on one side of the lagoon, whereas the radon data integrates the groundwater influence across the lagoon. Nevertheless, we have added the flying area to a map in our revised manuscript (green dotted lines, Figure 1) as we do think this is a good idea that will help the reader visualize the integrated data collection.*

**Sub-comment 2)** In lines 275-280, three springs were selected to determine the power function relationship between spring discharge and thermal plume area for the lagoon. There are about 40 springs in this area. Are the three springs representative? In addition, are the three data points too little to yield the mathematical relationship between the two?

*This is a very reasonable concern. Please see our response to reviewer 1, comment 9 to explain why more points are not possible. We have explained this in more detail in the revised text.*

**Sub-comment 3)** In line 281, the area of the spring is evaluated based on the irregular clipping of the spring location on the thermal image. What is the standard of graphic clipping? What principles need to be followed?

*There is no standard clipping approach in the literature, but the key point is to be consistent to allow comparison across the dataset.*

*Irregular clipping was conducted, where possible, to isolate two distinct thermal groups and the transition between them (the lagoon water, and the spring water). The main priority was to reduce interference from thermal groups with overlapping temperature ranges (e.g., foliage, shoreline). Rectangular cropping enables too many sources of thermal interference, which in several cases erroneously altered the calculated area. In general, adding more clipping information seems unnecessary as, if anything, we may have too much image processing information in this manuscript (see Reviewer 1 comment 3) and have reduced it.*

**Comment 8:** In line 253, the 1-D subsurface heat and water transport model established in the study area includes a saturated area of 3-93m. Do you have a temperature distribution along the perpendicular cross section to show the area that is effected by the spring plume. This is important to support that why the authors only select the temperature data at the depths of 1m, 3m, 5m, 10.28m, and 15.24m in the numerical modeling approach in response to the surface forcing (Fig. 10)?

*Please note we have flipped figures 9 and 10 to progress from seasonal discussions to climate change discussion. The only groundwater temperature data we have are in the coastal piezometer and upland well as described in the manuscript. We do not have data revealing the temperature distribution down to 93 m. We extend the model far below our depth of interest to remove any effects of geothermal heat flux as is common in such modeling, and we have explained this in the revised text (L319-320). The depths indicated here refer to standard depths to show damping and lagging (1, 3, and 5 m in what is now Fig. 9b). The other depths (10.28 and 15.24) in the original Figure 10b did give the appearance of being randomly selected. We*

*have removed the 15.24 m location and have changed 10.28 to 10.3 (exact depth was based on node spacing which didn't always land on exact metres) in what is now Figure 9b.*

**Comment 9:** In line 823, please change "Bottom row [(c) and (c)]" to " Bottom row[(c) and(d)]".

*Thank you for catching this typo, which has been fixed in the revision (Figure 4 caption).*

**Comment 10:** In Fig. 4(a)-4(b), please add the corresponding scale bar or pixel size of the image.

*Good point; this has been added to the Figure 4 caption.*

**Comment 11:** In Fig 10(a), the precipitation data over the years is unclear and lacks units. Please modify it.

*Thank you, somehow this must have got cut off during the image upload and manuscript compilation. We have fixed this in the revised version (now Figure 9).*

**Comment 12:** The work is comprehensive, it would be a good work if the main research goal and methods, especially their connections, can be stated very clear through the paper.

*Thank you. We feel that the modifications in response to concerns from both reviewers should help clarify the goals and methods of the study.*